# The lysosomal Ragulator complex plays an essential role in leukocyte trafficking by activating myosin II

Takeshi Nakatani[1,2,3,8], Kohei Tsujimoto[1,2,3,4,8], JeongHoon Park[4,5,8], Tatsunori Jo[1,2,3], Tetsuya Kimura[6], Yoshitomo Hayama[1,7], Hachiro Konaka[1,2,3,4], Takayoshi Morita[1,2,3], Yasuhiro Kato[1,2,3], Masayuki Nishide [1,2,3], Shyohei Koyama[1,2,3,4], Shigeyuki Nada[6], Masato Okada [6], Hyota Takamatsu [1,2,3,4 ✉] & Atsushi Kumanogoh [1,2,3 ✉]

Lysosomes are involved in nutrient sensing via the mechanistic target of rapamycin complex 1 (mTORC1). mTORC1 is tethered to lysosomes by the Ragulator complex, a heteropentamer in which Lamtor1 wraps around Lamtor2–5. Although the Ragulator complex is required for cell migration, the mechanisms by which it participates in cell motility remain unknown. Here, we show that lysosomes move to the uropod in motile cells, providing the platform where Lamtor1 interacts with the myosin phosphatase Rho-interacting protein (MPRIP) independently of mTORC1 and interferes with the interaction between MPRIP and MYPT1, a subunit of myosin light chain phosphatase (MLCP), thereby increasing myosin II–mediated actomyosin contraction. Additionally, formation of the complete Ragulator complex is required for leukocyte migration and pathophysiological immune responses. Together, our findings demonstrate that the lysosomal Ragulator complex plays an essential role in leukocyte migration by activating myosin II through interacting with MPRIP.

[1] Department of Respiratory Medicine and Clinical Immunology, Graduate School of Medicine, Osaka University, Suita, Osaka, Japan. [2] Department of Immunopathology, WPI, Immunology Frontier Research Center (iFReC), Osaka University, Suita, Osaka, Japan. [3] Integrated Frontier Research for Medical Science Division, Institute for Open and Transdisciplinary Research Initiatives, Osaka University, Suita, Osaka, Japan. [4] The Japan Science and Technology – Core Research for Evolutional Science and Technology (JST–CREST), Osaka University, Suita, Osaka, Japan. [5] Department of Internal Medicine, Osaka Police Hospital, Ten-nouji-Ku, Osaka, Japan. [6] Department of Oncogene Research, Research Institute for Microbial Diseases, Osaka University, Suita, Osaka, Japan. [7] Department of Respiratory Medicine, Kinki Central Hospital, Itami, Hyogo, Japan. [8] These authors contributed equally: Takeshi Nakatani, Kohei Tsujimoto, JeongHoon Park. ✉email: thyota@imed3.med.osaka-u.ac.jp; kumanogo@imed3.med.osaka-u.ac.jp

Lysosomes are membrane-bound organelles that degrade macromolecules to recycle their constituent metabolites. Dysfunction in this process causes lysosomal storage disorders and primary immunodeficiency syndromes[1, 2]. Recent work showed that lysosomes function as nutrient-sensing signaling platforms where mTORC1 binds the Ragulator–Rag GTPase complex[3]. The Ragulator complex is a pentamer containing two heterodimers: Lamtor1/p18 wraps around Lamtor2/p14–Lamtor3/MP1 and Lamtor4/p10–Lamtor5/HBXIP and anchors the Ragulator complex to the lysosomal membrane, where Ragulator, in turn, tethers the Rag GTPase heterodimers RagA/B and RagC/D[4, 5]. Under nutrient-rich conditions, mTORC1 is immobilized to the lysosomal membrane via the Ragulator complex, leading to activation downstream signaling factors such as p70S6 kinase 1 (S6K) that promote protein synthesis, cell proliferation, and autophagy in various cell types[6, 7]. The Ragulator complex is also expressed in immune cells, and deficiency of Lamtor1 in macrophages results in mTORC1-dependent impairment of M2 macrophage polarization due to aberrant lipid metabolism[8]. The lack of Lamtor1 leads to reduced phosphorylation of S6K and lower levels of the LXR ligand, 25-hydroxycholesterol, resulting in the expression of LXR-dependent genes that are important for the differentiation of M2 macrophages. Also, pro-inflammatory cytokine production is elevated in *Lamtor1*−/− macrophages due to the increase in nuclear translocation of transcription factor EB (TFEB)[9].

Lysosomes and their positioning have been shown to be involved in cell motility[10, 11]. In leukocytes, specialized motile cells that perform immune surveillance[12], lysosomes transported to the uropod also promote cell motility by providing $Ca^{2+}$ to facilitate cytoskeletal organization[13], as well as by exocytosing their contents to degrade the extracellular matrix (ECM), allowing cells to detach their trailing edge from the ECM[14]. Rather than using integrins, leukocytes use actomyosin contraction, especially in a three-dimensional (3D) environment[15, 16]. Dendritic cells (DCs), professional antigen-presenting leukocytes that bridge innate and acquired immunity by delivering peripheral antigens to secondary lymphoid organs in order to promote antigen-specific T-cell priming[17, 18], can traverse across many physical barriers by adopting migratory modes suitable for 2D or 3D environments[19, 20]. In confined 3D environments, they predominantly use the actomyosin contractile force generated by the activation of myosin II[15, 16, 19]. Phosphorylation of the myosin light chain (MLC) is regulated by the balance between $Ca^{2+}$/calmodulin-mediated MLC kinase (MLCK) and MLC phosphatase (MLCP). MLCP is a heterotrimer consisting of myosin phosphatase-targeting subunit 1 (MYPT1), a 20-kDa small subunit (M20), and a catalytic subunit of the type I protein serine/threonine phosphatase family (PP1cδ)[16]. MLCP is anchored on actin–myosin bundles by myosin phosphatase-Rho interacting protein (MPRIP, also known as p116$^{Rip}$)[21], and MLCP activity is regulated by RhoA via suppression of MYPT1 phosphorylation, which inactivates the catalytic activity of PP1cδ[22]. However, the molecular mechanism of myosin II activation via lysosomes in DCs has not been completely elucidated.

In this study, we show that Ragulator complex expressed on the lysosomal membrane plays a critical role in leukocyte migration by regulating myosin II-dependent actomyosin contraction. During DC locomotion, lysosomes localize to the uropod, where the Ragulator complex interacts with MPRIP; this interferes with the interaction between MPRIP and MYPT1, leading to decreased MLCP activity. Additionally, a mutant of Lamtor1 that is unable to form the Ragulator complex fails to restore MLC phosphorylation and cell migration due to defects in MPRIP interaction. Furthermore, DC- and neutrophil-specific Lamtor1-deficient mice have reduced numbers of migratory DC subsets in skin

dLNs and infiltrated neutrophils in inflamed peritoneum due to impairment of in vivo migration, resulting in pathogenic immune responses.

## Results

**Lysosome-expressed Lamtor1 localizes to the uropod during DC migration**. We first visualized lysosomes with AcidiFluor ORANGE[23], which becomes more fluorescent in acidic organelles, to determine the localization of lysosomes during DC migration. In this experiment, we examined DC locomotion within a 2 mg/ml type I collagen matrix that mimicked an in vivo 3D environment[15, 24]. Consistent with previous reports[15, 25], the cell bodies of migrating DCs adopted a stretched shape due to protrusion of the leading edge. Subsequently, the cell bodies were pushed forward due to the contraction of the uropod. In the contraction phase, AcidiFluor-positive lysosomes were localized to the cell body, as well as in the rear of the elongated DCs (Fig. 1a, Supplementary movie 1). Additionally, lysosomes detected by the anti-LAMP1 antibody were preferentially localized to the perinuclear regions of non-polarized cells, but to the peripheral regions and uropod of polarized cells (Fig. 1b). Given that Ragulator complex expressed on the lysosomal membrane is involved in cell movement[26, 27], we focused on Lamtor1 and examined the localization of Lamtor1 in polarized and non-polarized cells. The distribution of Lamtor1 protein resembled that of Lamp1 (Fig. 1c), and Lamtor1 and Lamp1 were co-localized regardless of cell polarization (Fig. 1d). Furthermore, phosphorylated MLC was present on both the leading edge and uropod of polarized cells, whereas Lamtor1 was co-localized with phosphorylated MLC in the uropod (Fig. 1e). Therefore, we hypothesized that Lamtor1 expressed on the lysosomal membrane regulates cell motility by regulating myosin II-mediated actomyosin contraction in the uropod.

**Failure of DC migration in a 3D environment under Lamtor1 deficiency**. To evaluate the involvement of Lamtor1 in DC migration, we generated DC-specific *Lamtor1*-deficient mice (CD11c–*Lamtor1*$^{flox/flox}$) by crossing *Lamtor1*$^{flox/flox}$ mice with CD11c-Cre mice (Supplementary Fig. 1) and performed a chemotaxis assay using Transwell. The response of *Lamtor1*−/− bone marrow-derived DCs (BMDCs) to CCL19 or CCL21 was lower than that of WT DCs, although the expression levels of CCR7 were comparable (Fig. 2a, Supplementary Fig. 2a). Analysis of single-cell motilities by time-lapse video imaging revealed that the mean velocity of random DC migration on fibronectin-coated coverslips and the directionality and velocity of DCs in response to CCL19 was comparable between WT and *Lamtor1*−/− DCs (Supplementary Fig. 2b–d, Supplementary Movies 2, 3). The percentage of adherent cells on fibronectin-coated plates did not differ between WT and *Lamtor1*−/− DCs (Supplementary Fig. 2e). On the other hand, *Lamtor1*−/− DCs that migrate in response to CCL19 in a 3D collagen matrix consisting of 2 mg/ml Type I collagen to mimic the physiological 3D environment of DCs, formed an elongated shape due to impaired uropod retraction and normal protrusion of the leading edge, resulting in impaired DC migration (Fig. 2b–f, Supplementary Movies 4, 5). Additionally, MLC phosphorylation was lower in *Lamtor1*−/− DCs than in WT DCs even under non-stimulated conditions (Fig. 2g), suggesting that Lamtor1 is required for DC motility in 3D, but not 2D, environments and that Lamtor1 regulates myosin II activity. Furthermore, lysosomes tended to be distributed to the peripheral region in non-polarized *Lamtor1*−/− DCs, as observed previously in fibroblasts[26, 28], and also localized at the uropod of polarized *Lamtor1*−/− DCs (Fig. 2h). The expression level of Lamp1 protein in *Lamtor1*−/− DCs was comparable to that in

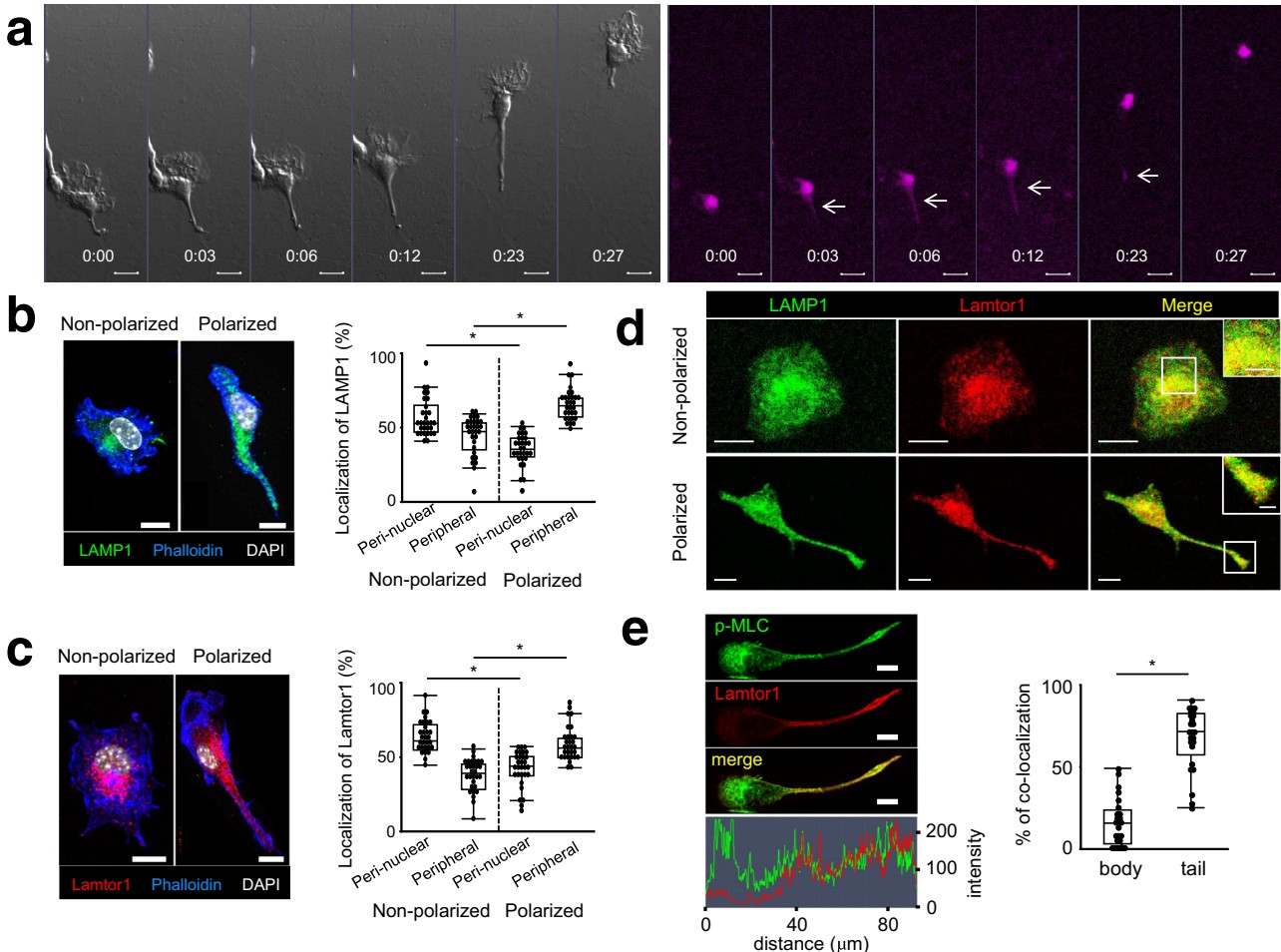

**Fig. 1 Localization of the lysosome and Lamtor1 to the rear of migrating DCs. a** Localization of lysosomes during DC migration in a confined 3D environment. BMDCs were labeled with 1 μM AcidiFluor ORANGE and stimulated with LPS for 2 h. Movement of DCs (left) and AcidiFluor-positive lysosomes (magenta) in response to CCL19 (5 μg/ml) in type I collagen gel (2 mg/ml) in a Zigmond chamber were observed at 1-min intervals by time-lapse video imaging. **b, c** Localization of LAMP1+ lysosomes (**b**) and Lamtor1 (**c**) in non-polarized and polarized DCs. WT DCs were stained with anti-LAMP1 (green) (**b**), anti-Lamtor1 (red) (**c**), phalloidin (blue), and DAPI (white), and then visualized by confocal microscopy. Polarized and non-polarized cells were identified morphologically. Representative images are shown. Scale bar, 10 μm (left). The percentages of the ROI in the perinuclear and peripheral regions were determined (n = 30) (right). The perinuclear region was defined as the region within 5 μm of the nuclear membrane, and the peripheral region was defined as the region of the cell outside the perinuclear region. **d** Co-localization of Lamtor1 and LAMP1 in non-polarized (upper) and polarized (lower) DCs. DCs were stained with anti-Lamtor1 (red), anti-LAMP1 (green), and DAPI (white), and then visualized by confocal microscopy. Representative images are shown. Scale bar, 10 μm. Data are representative of three experiments. **e** Co-localization of Lamtor1 and phosphorylated MLC. Lamtor1 and phosphorylated MLC were visualized by staining in THP1 expressing Lamtor1-FLAG (red) with anti-p-MLC (green). Representative confocal images of polarized cells (left); scale bar, 10 μm (upper). Intensities of Lamtor1 (red) and p-MLC (green) (lower). Percentage of co-localization of Lamtor1 and p-MLC in body or tail region of polarized cells (n = 25 cells) (right). Data are representative of three experiments. Statistical analyses were performed by two-sided Steel–Dwass test (**b, c**) or Mann–Whitney U test (**e**) [median; 25th and 75th percentiles; and minimum and maximum of a population excluding outliers; *p < 0.0001].

WT DCs (Fig. 2i). These results suggest that impaired DC migration was not due to impairment of lysosome generation or distribution to the uropod.

**DCs migrate independently of mTORC1.** Lamtor1 is a scaffold protein of mTORC1[7]. Several studies have reported that rapamycin, a mTORC1-specific inhibitor, interferes with leukocyte trafficking, although these findings remain somewhat controversial[29–32]. When DCs were treated with rapamycin and torin1 (a mTORC1/mTORC2 inhibitor) at doses that completely inhibited phosphorylation of S6K, DC motility in response to CCL19 was not affected (Fig. 3a, b), regardless of amino acid concentration (Supplementary Fig. 3a, b). Additionally, the motility and morphology of migrating DCs in the 3D collagen matrix were similar (Fig. 3c), and the level of MLC

phosphorylation was also comparable between rapamycin-treated and rapamycin-untreated DCs (Fig. 3d). Interestingly, phosphorylation of S6K was observed in splenic DCs and BMDCs, irrespective of the presence of Lamtor1 (Supplementary Fig. 3c, d). These results suggested that S6K phosphorylation is dispensable for DC migration and that Lamtor1 is not essential for S6K phosphorylation in DCs. In addition, the Ragulator complex is involved in the MEK pathway because the Lamtor2/3 complex is a scaffold for MEK[26, 33]. However, DCs treated with the MEK inhibitor U0126 did not decrease CCL19-induced cell motility even at high drug concentrations (Fig. 3e). Moreover, phosphorylation of MLC did not differ between U0126-treated DCs and untreated DCs (Fig. 3f). These results suggested that the MEK-dependent pathway is dispensable for DC migration.

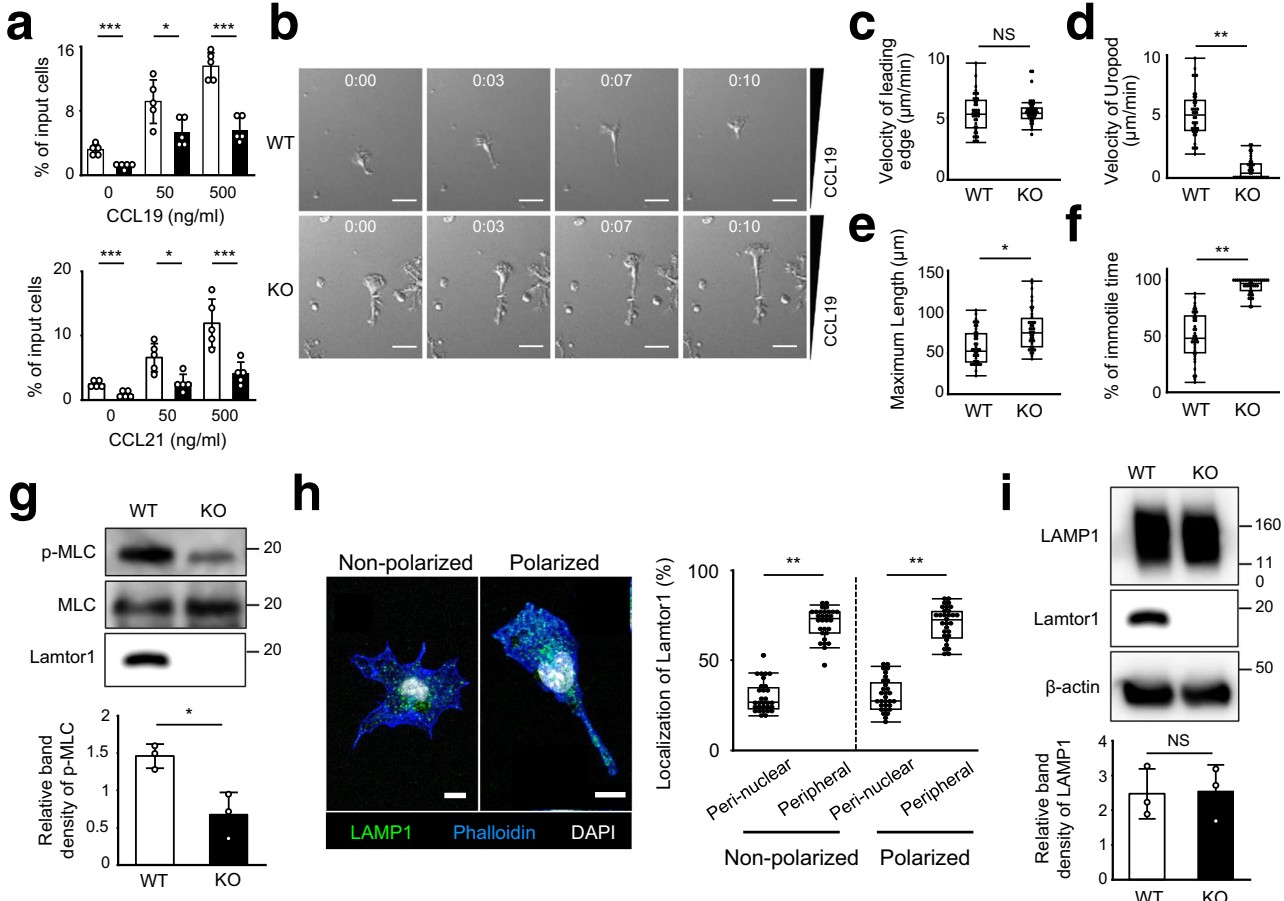

**Fig. 2 Impairment of DC migration in constricted areas under Lamtor1 deficiency. a** Chemotaxis of WT (white bar) and *Lamtor1*$^{-/-}$ (black bar) BMDCs toward the indicated concentrations of CCL19 (upper) or CCL21 (lower) in a Transwell assay system (pore size, 5 μm). Data are representative of three experiments. **b–f** Motility of DCs in response to CCL19 in 3D collagen matrices. Movement of WT (upper) and *Lamtor1*$^{-/-}$ (lower) BMDCs in response to CCL19 (5 μg/ml) in type I collagen gels (2 mg/ml) in a Zigmond chamber were observed at 1-min intervals by time-lapse video imaging. Velocities of DCs were determined using the ImageJ manual tracking software. Consecutive images of DC locomotion: time (h:min) is shown above each panel. Scale bar, 30 μm (**b**). Mean velocity of the leading edge (**c**) and uropod (**d**), the maximum length of a single cell (**e**), and percentage of the immotile period of the uropod (**f**) (n = 40 WT cells and n = 40 KO cells). **g** MLC phosphorylation in WT and *Lamtor1*$^{-/-}$ BMDCs. MLC phosphorylation was evaluated by western blotting with anti-p-MLC antibody (upper). Data are representative of three experiments. Protein concentration in SDS–PAGE gel bands was determined using ImageJ, and statistical analysis was performed (lower) (n = 3). **h** Localization of LAMP1$^+$ lysosomes in non-polarized and polarized DCs. *Lamtor1*$^{-/-}$ DCs were stained with anti-LAMP1 (green), phalloidin (blue), and DAPI (white), and visualized by confocal microscopy. Representative images of a non-polarized cell (left) and a polarized cell (right). Scale bar, 10 μm (left). Percentage of the region of interest (ROI) of perinuclear and peripheral regions of non-polarized cells and polarized cells (n = 30) (right). **i** Lamp1 protein level of WT and *Lamtor1*$^{-/-}$ BMDCs. The amount of Lamp1 protein was evaluated by western blotting with anti-Lamp1 antibody (upper). Data are representative of three experiments. The protein concentration in SDS–PAGE gel bands was determined using ImageJ and statistical analysis was performed (lower) (n = 3). Statistical analyses were performed by two-sided Student's *t*-test (**a**, **g**, **i**) [means ± s.d.; *p < 0.05, **p < 0.01, ***p < 0.005] or two-sided Mann–Whitney *U* test (**c–f**, **h**) [median; 25th and 75th percentiles; and minimum and maximum of a population excluding outliers; *p < 0.001, **p < 0.0001; NS, not statistically significant].

**Lamtor1 interacts with MPRIP independently of mTOR.** Next, in order to identify the Lamtor1-associated molecules, we performed LC–MS analysis of lysates of WT and *Lamtor1*$^{-/-}$ BMDCs following pull-down with an anti-Lamtor1 antibody. Component proteins of the Ragulator complex, Lamtor2, were pulled down (Supplementary Table 1). We focused on the MPRIP because it has been shown to anchor MYPT1, an MLCP subunit, on the actin–myosin bundle to facilitate MLCP activity[34, 35]. To determine whether Lamtor1 interacts with MPRIP, we established HEK293T cells stably expressing V5-tagged MPRIP (MPRIP-V5-HEK293T) and co-expressed FLAG-tagged Lamtor1 (Lamtor1-FLAG) in these cells. In immunoprecipitation experiments, Lamtor1 and MPRIP co-precipitated with each other (Fig. 4a). In addition, Lamtor1 and MPRIP co-localized to the uropod (Fig. 4b). Furthermore, the interaction between Lamtor1 and

MPRIP occurred regardless of rapamycin treatment (Fig. 4c). These results suggest that Lamtor1 and MPRIP interact with each other in the uropod independently of mTORC1.

**Interaction of Lamtor1 with MPRIP interferes with the binding between MPRIP and MYPT1.** To further investigate whether Lamtor1 interferes with the binding of MPRIP to MYPT1, we first established a Lamtor1-deficient derivative of the human monocyte-like cell line THP1, *Lamtor1*-KO-THP1, using the CRISPR/Cas9 system. We then re-expressed FLAG-tagged full-length Lamtor1 in these cells (*Lamtor1*-KO-Full-THP1) (Supplementary Fig. 4). We investigated the interaction between MPRIP and MYPT1 by immunoprecipitating parental WT-THP1, *Lamtor1*-KO-THP1, and *Lamtor1*-KO-Full-THP1 with

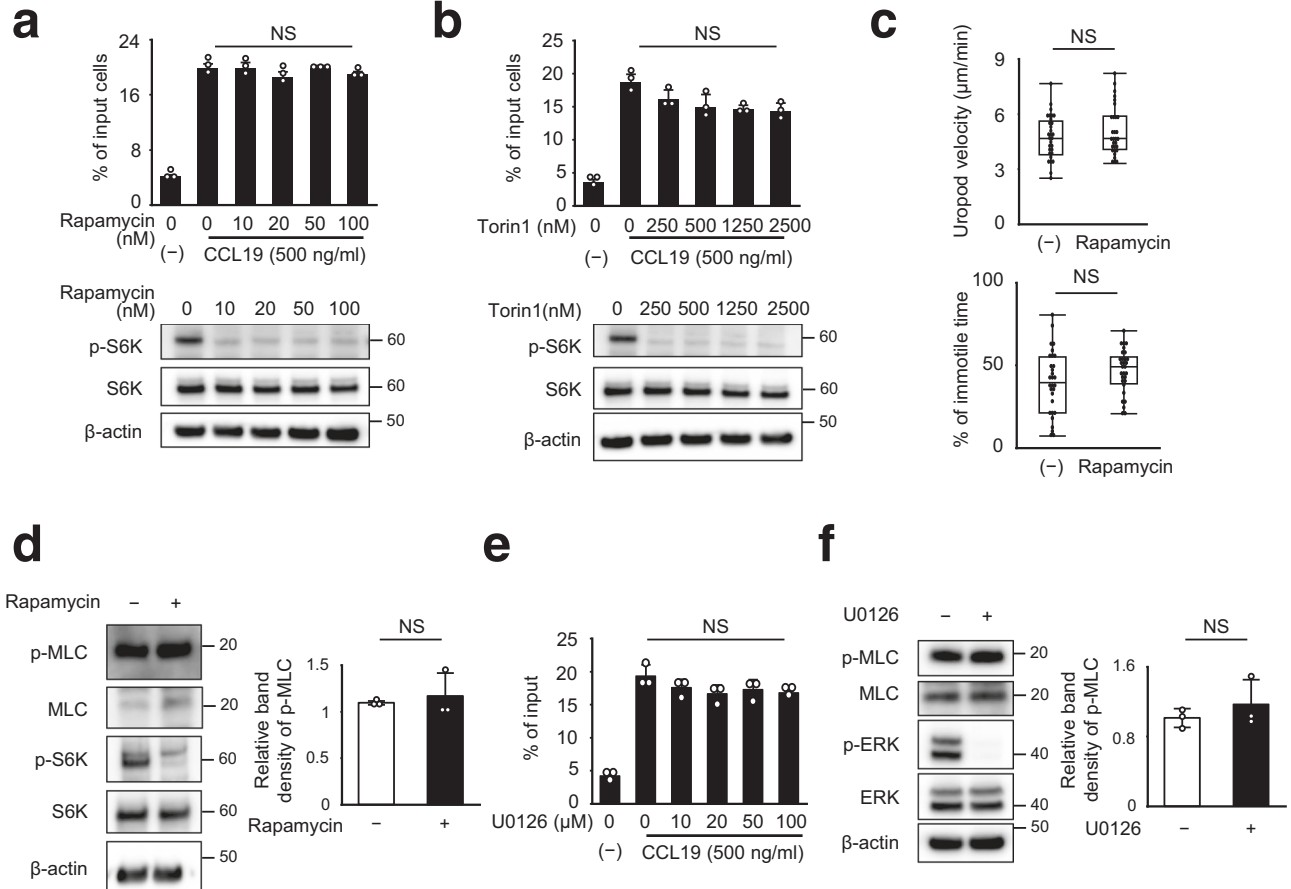

**Fig. 3 mTOR inhibition does not affect DC migration. a, b** Chemotaxis of DCs treated with mTORC1 inhibitors in response to CCL19. Chemotaxis of WT BMDCs in response to CCL19 (500 ng/ml) in the presence of the indicated concentration of rapamycin (**a**) and torin1 (**b**), as determined by Transwell assay (pore size, 5 μm) (upper). Data are representative of three experiments (mean ± s.d.). Phosphorylation of S6K following treatment with rapamycin (**a**) and Torin1 (**b**) was determined by western blotting (lower). **c** DC migration in 3D collagen matrices in the presence of rapamycin. Movement of WT BMDCs in response to CCL19 (5 μg/ml) in type I collagen gel (2 mg/ml) containing 10 nM rapamycin was observed by time-lapse video imaging, and the velocities of the uropod (upper) and the percentage of the immotile period of the uropod (lower) were analyzed using the ImageJ manual tracking software ($n = 40$). **d** MLC phosphorylation of DCs treated with rapamycin. MLC phosphorylation and S6K phosphorylation in the presence of 10 nM rapamycin was evaluated by western blotting with anti-p-MLC and anti-p-S6K antibody, respectively (left). Data are representative of three experiments. The concentration of p-MLC in SDS-PAGE gel bands was determined using ImageJ, and statistical analysis was performed (right) ($n = 3$). **e** Chemotaxis of U0126-treated DCs in response to CCL19. Chemotaxis of WT BMDCs in response to CCL19 (500 ng/ml) in the presence of the indicated concentration of U0126, a MEK inhibitor, as determined by Transwell assay (pore size, 5 μm) (upper). Data are representative of three experiments (mean ± s.d.). **f** MLC phosphorylation of DCs treated with U0126. MLC and ERK phosphorylation in the presence of 20 μM U0126 were determined by western blotting with anti-p-MLC and anti-p-ERK antibody (left). Data are representative of three experiments. The concentration of p-MLC in SDS-PAGE gel bands was determined using ImageJ, and statistical analysis was performed (right) ($n = 3$). Statistical analysis was performed by two-sided Student's $t$-test (**a**, **b**, **d-f**) [means ± s.d.], and two-sided Mann-Whitney $U$ test (**c**) [median; 25th and 75th percentiles; and minimum and a maximum of a population; NS not statistically significant].

anti-MPRIP antibody. The co-precipitation of MYPT1 and MPRIP was lower in WT-THP1, but higher in *Lamtor1*-KO-THP1, whereas the increase in co-precipitation in *Lamtor1*-KO-THP1 was reduced in *Lamtor1*-KO-Full-THP1 (Fig. 5a). In addition, we compared the localization of MPRIP and MYPT1 in WT and Lamtor1-KO THP1 cells by confocal microscopy. Interestingly, in WT THP1, localization of MYPT1 to the uropod was reduced, whereas in *Lamtor1*-KO THP1, MYPT1 was present in the uropod, and it co-localized with MPRIP in the uropod to a greater extent than in WT THP1 (Fig. 5b). We further evaluated the importance of the Lamtor1–MPRIP interaction in cell migration by knocking down endogenous MPRIP in Lamtor1-KO-THP1. MLC phosphorylation and chemotaxis, which were reduced in *Lamtor1*-KO-THP1, were restored to the same level as in WT-THP1 by knockdown of MPRIP (Fig. 5c). These results indicate that the interaction between Lamtor1 and MPRIP is responsible for cell motility.

**Lamtor1-mediated motility is driven by myosin II activity.** Next, we investigated whether myosin II-dependent actomyosin is important for Lamtor1-mediated motility. MLC phosphorylation, which was reduced in *Lamtor1*-KO-THP1, was restored to the same level as in WT-THP1 by add-back of Lamtor1 (Fig. 6a). In addition, chemotaxis in response to CCL2 was reduced in *Lamtor1*-KO-THP1, yet restored to the same level as in WT-THP1 in *Lamtor1*-KO-Full-THP1, and the restored motility in *Lamtor1*-KO-Full-THP1 was diminished by treatment with blebbistatin, a myosin II inhibitor (Fig. 6b). Furthermore, motility, which was reduced in *Lamtor1*-KO THP1, was restored by forced expression of a constitutively active form of MLC (MLC-DD, carrying the Thr18Asp and Ser19Asp mutations)[36] in *Lamtor1*-KO-THP1 (Fig. 6c). On the other hand, the amount of GTP-bound RhoA and the phosphorylation level of MYPT1 were comparable between WT and *Lamtor1*$^{-/-}$ DCs, although MPRIP has been shown to affect the phosphorylation of MYPT1 by

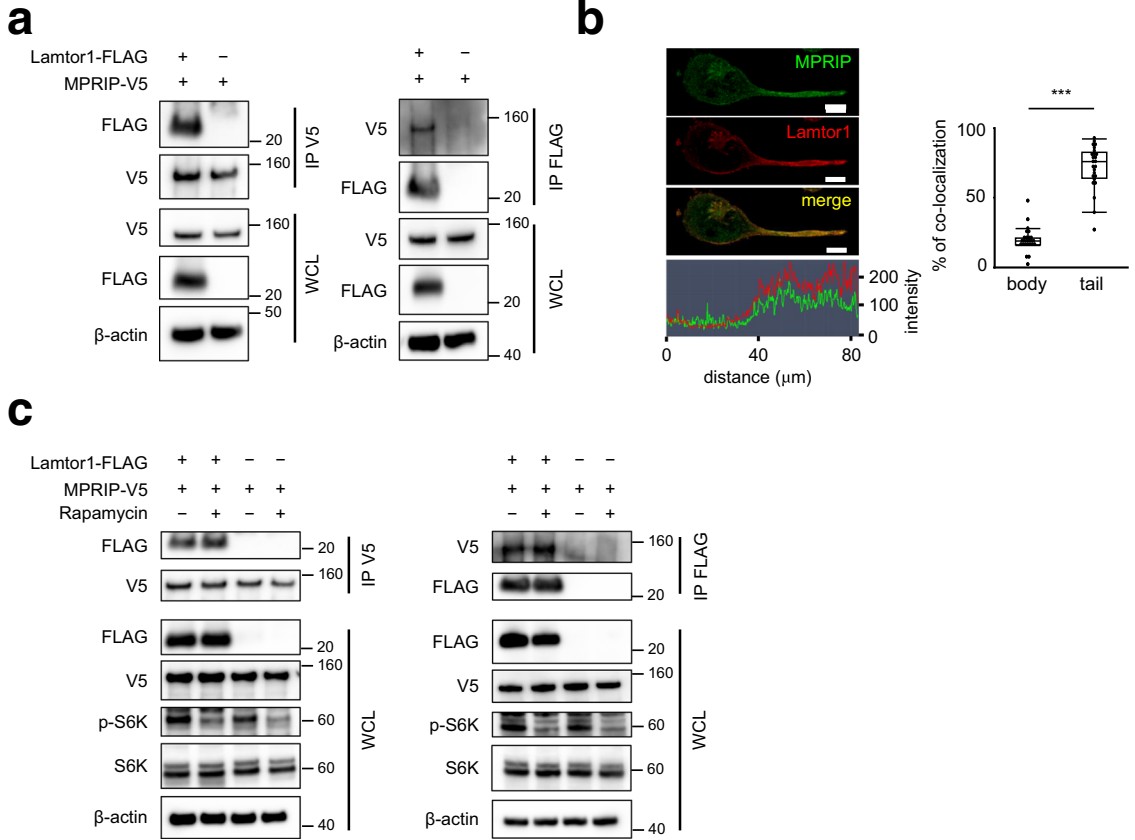

**Fig. 4 Lamtor1 interacts with MPRIP independently of mTORC1. a** Interacting of Lamtor1 and MPRIP. FLAG-tagged Lamtor1 expression vector or control vector was transfected into V5-tagged-MPRIP-expressing HEK293T cells. Cells were lysed, precipitated with anti-V5 antibody (left) or anti-FLAG antibody (right), and detected by western blotting with anti-V5 and -FLAG antibodies. Whole-cell lysates (WCL) were detected with anti-β-actin antibody. Data are representative of three experiments. **b** Co-localization of Lamtor1 and MPRIP. Localization of MPRIP in Lamtor1-FLAG-expressing THP1 was evaluated by confocal microscopy using anti-MPRIP and anti-FLAG antibodies. A representative image of MPRIP (green) and Lamtor1 (red) in a polarized cell is shown. Scale bar, 10 μm (upper), and the intensity of MPRIP (green) and Lamtor1 (red) signals (bottom) (left). Percentage of co-localization of Lamtor1 and MPRIP in the body or tail region of polarized cells (*n* = 40) (right). **c** Lamtor1 interacts with MPRIP independent of mTORC1. Forty-eight hours after transfection of FLAG-Lamtor1 expression vector or control vector into V5-MPRIP-expressing HEK293T cells, the cells were treated with 10 nM rapamycin for 2 h. The lysates were precipitated with anti-V5 (left) or anti-FLAG antibody (right), and the immunoprecipitates were subjected to western blotting with anti-V5 and -FLAG antibodies. WCL was blotted with anti-β-actin, anti-S6K, and anti-phospho-S6K antibodies. Data are representative of three experiments. Statistical analyses were performed by two-sided Mann–Whitney *U* test (**b**) [median; 25th and 75th percentiles; and minimum and maximum of a population excluding outliers; ***$p < 0.0001$].

inactivating the RhoA–ROCK pathway[35] (Fig. 6d, e). These results suggest that Lamtor1-mediated cell motility is driven by myosin II activity induced by Lamtor1's interference with the MPRIP–MYPT1 interaction, but not by activation of the RhoA–ROCK pathway.

**Ragulator complex formation and interacting with MPRIP are necessary for cell motility.** Lamtor1 is a component of the heteropentameric Ragulator complex[4, 5]. Thus, to determine the regions of Lamtor1 that are critical for the assembly of the Ragulator complex formation and interaction with MPRIP, we divided Lamtor1 into four parts based on the structure of the Ragulator complex[4]: the C-terminal tail of Lamtor1, the part facing the cytoplasm side (α3–α4 helix), the part facing the lysosome side (α1–α2 helix), and the part near the stalk to anchor the Ragulator complex to the lysosome (the N-terminal–α1 helix) (Supplementary Fig. 5a). We generated three FLAG-tagged truncated forms of Lamtor1; ΔT1 (Met1–Ser144), ΔT2 (Met1–Gln114), and ΔT3 (Met1–His41) (Supplementary Fig. 5b). All truncated Lamtor1 variants failed to form the Ragulator complex (Supplementary Fig. 5c), confirming that the C-terminal

tail of Lamtor1 is essential for Ragulator complex formation. When we introduced the ΔT1-mutant into MPRIP-V5-HEK293T cells, the interaction of MPRIP with ΔT1 mutant was completely abolished (Fig. 7a, b). Because the α4E mutant (in which three amino acids in the α4 helices were replaced by glutamic acid)[4] fails to form the Ragulator complex, we transduced the FLAG-tagged mutant of α4E-Lamtor1 into MPRIP-V5-HEK293T cells to determine whether the C-terminal region of Lamtor1 or Ragulator complex formation is important for interacting with MPRIP. Interestingly, we did not observe an interaction between MPRIP and the α4E mutant (Fig. 7a, c), suggesting that Ragulator complex formation is required for interacting with MPRIP, as the α4E mutant has an intact C-terminal tail and the ΔT1-mutant has an intact α4 helix region (Fig. 7a). Indeed, the protein levels of Lamtor2–5 were reduced in *Lamtor1*−/− DCs, *Lamtor1*-KO-THP1, and *Lamtor1*-KO-THP1 expressing mutant forms of Lamtor1 (Supplementary Fig. 6a, b), implying that each component of the complex is required for the stability of the others. Furthermore, the enhanced interaction between MPRIP and MYPT1, as well as MLC phosphorylation in *Lamtor1*-KO-THP1, was restored in *Lamtor1*-KO-Full-THP1 but not in *Lamtor1*-KO-

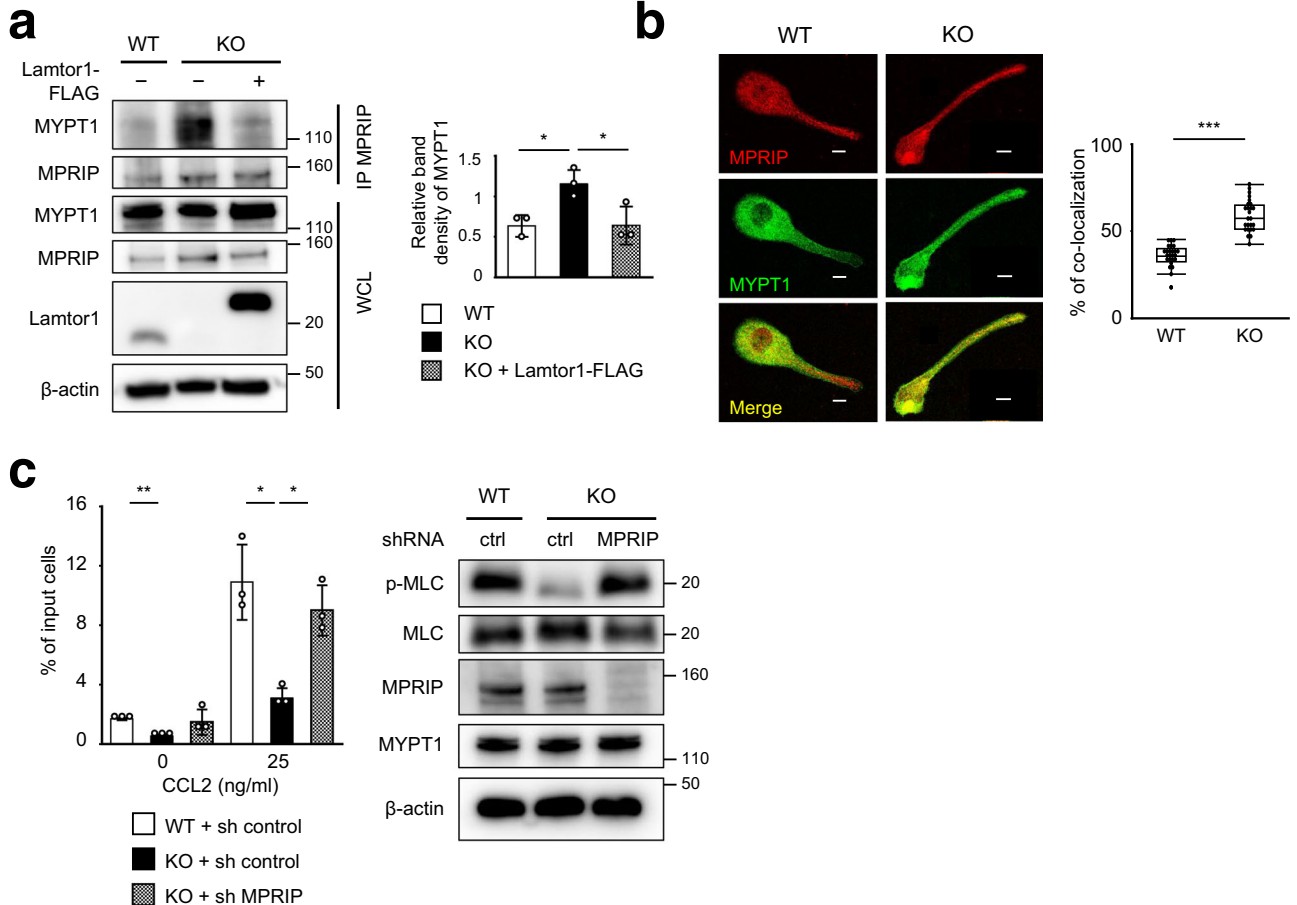

**Fig. 5 Interacting of Lamtor1 with MPRIP interferes with the interaction between MPRIP and MYPT1. a** Interacting of MPRIP and MYPT1 in the presence or absence of Lamtor1. Parental THP1 cells (WT-THP1), Lamtor1-deficient THP1 (*Lamtor1*-KO-THP1), and *Lamtor1*-KO-THP1 cells in which full-length Lamtor1-FLAG was re-expressed (*Lamtor1*-KO-Full-THP1) were lysed, precipitated with anti-MPRIP antibody, and detected by western blotting with anti-MYPT1 and anti-MPRIP antibodies (left). Data are representative of three experiments. Concentration of MYPT1 co-precipitated with MPRIP in SDS–PAGE gel bands were determined using ImageJ, and statistical analysis was performed (right). **b** Co-localization of MPRIP and MYPT1. Localization of MPRIP and MYPT1 in WT and *Lamtor1*-KO-THP1 was evaluated by confocal microscopy using anti-MPRIP and anti-MYPT1 antibodies. A representative image of MYPT1 (green) and MPRIP (red) in a polarized cell is shown. Scale bar, 10 μm (left). Percentage of co-localization of MPRIP and MYPT1 in the tail region of polarized cells ($n = 40$) (right). **c** MLC phosphorylation and chemotaxis of MPRIP-knockdown THP1 cells. WT-THP1 or *Lamtor1*-KO-THP1 cells were transduced with control shRNA (WT-sh-ctrl-THP1, *Lamtor1*-KO-sh-ctrl-THP1) or MPRIP-shRNA (*Lamtor1*-KO-sh-MPRIP-THP1). Cells were lysed, and the levels of MPRIP, MYPT1, and MLC phosphorylation were detected by western blotting (right). Chemotaxis of WT-sh-ctrl-THP1 (white bar), Lamtor1-KO-sh-ctrl-THP1 (black bar), and *Lamtor1*-KO-sh-MPRIP-THP1 (mesh bar) in response to CCL2 (25 ng/ml) was evaluated by Transwell assay (pore size, 5 μm) (left). Data are representative of three experiments. Statistical analysis was performed by two-sided ANOVA with Tukey's post hoc test (**a**, **c**) [means ± s. d.; *$p < 0.05$, **$p < 0.01$], or two-sided Mann–Whitney *U* test (**b**) [median; 25th and 75th percentiles; and minimum and maximum of a population excluding outliers; ***$p < 0.0001$].

α4E-THP1 (Fig. 7d, e). Furthermore, *Lamtor1*-KO-Full-THP1 moved as much as WT-THP1 in response to CCL2, whereas neither *Lamtor1*-KO-α4E-THP1 nor *Lamtor1*-KO-ΔT1-THP1 moved (Fig. 7f). Taken together, these results indicated that Ragulator complex formation plays an essential role in cell migration by modulating myosin II activity, which it achieves by interfering with the interaction between MPRIP and MYPT1.

**Impaired DC trafficking in *Lamtor1*⁻/⁻ DCs.** We then assessed the physiological significance of the Ragulator complex in DC-trafficking and immunological responses. CD11c–*Lamtor1*^flox/flox mice had a reduced number of CD11c⁺MHCII^hi DCs and CD11c⁺CD103⁺DCs, both of which are representative of migratory DCs, in skin-draining LNs (dLNs)[20, 37] (Fig. 8a), although they had abundant DCs in the skin itself (Fig. 8b). Additionally, we performed a DC mobilization experiment by painting FITC-

isomer onto the back skin[38]. The number of FITC-positive DCs in skin-dLNs after FITC skin sensitization was reduced in CD11c–*Lamtor1*^flox/flox mice (Fig. 8c). We then performed subcutaneous injections of Evans blue dye to evaluate the lymphatic flow system and rule out the possibility of impaired delivery of FITC-isomers. In addition, we performed the FITC microbead injection assay to evaluate the ability of in vivo DC migration more strictly, as microbeads cannot be delivered by the lymphatic flow. CD11c–*Lamtor1*^flox/flox mice exhibited no anatomical abnormality of the lymphatic system (Supplementary Fig. 7a), and had a lower number of FITC-microbead-bearing DCs in dLNs than WT mice (Fig. 8d). *Lamtor1*⁻/⁻ DCs also exhibit FITC-dextran uptake comparable to that of WT cells (Supplementary Fig. 7b). Furthermore, we performed subcutaneous injection of CFSE-labeled WT or *Lamtor1*⁻/⁻ DCs into WT footpads. CFSE-positive *Lamtor1*⁻/⁻ DCs were less abundant in dLNs 48 h after injection of CFSE-labeled DCs (Fig. 8e).

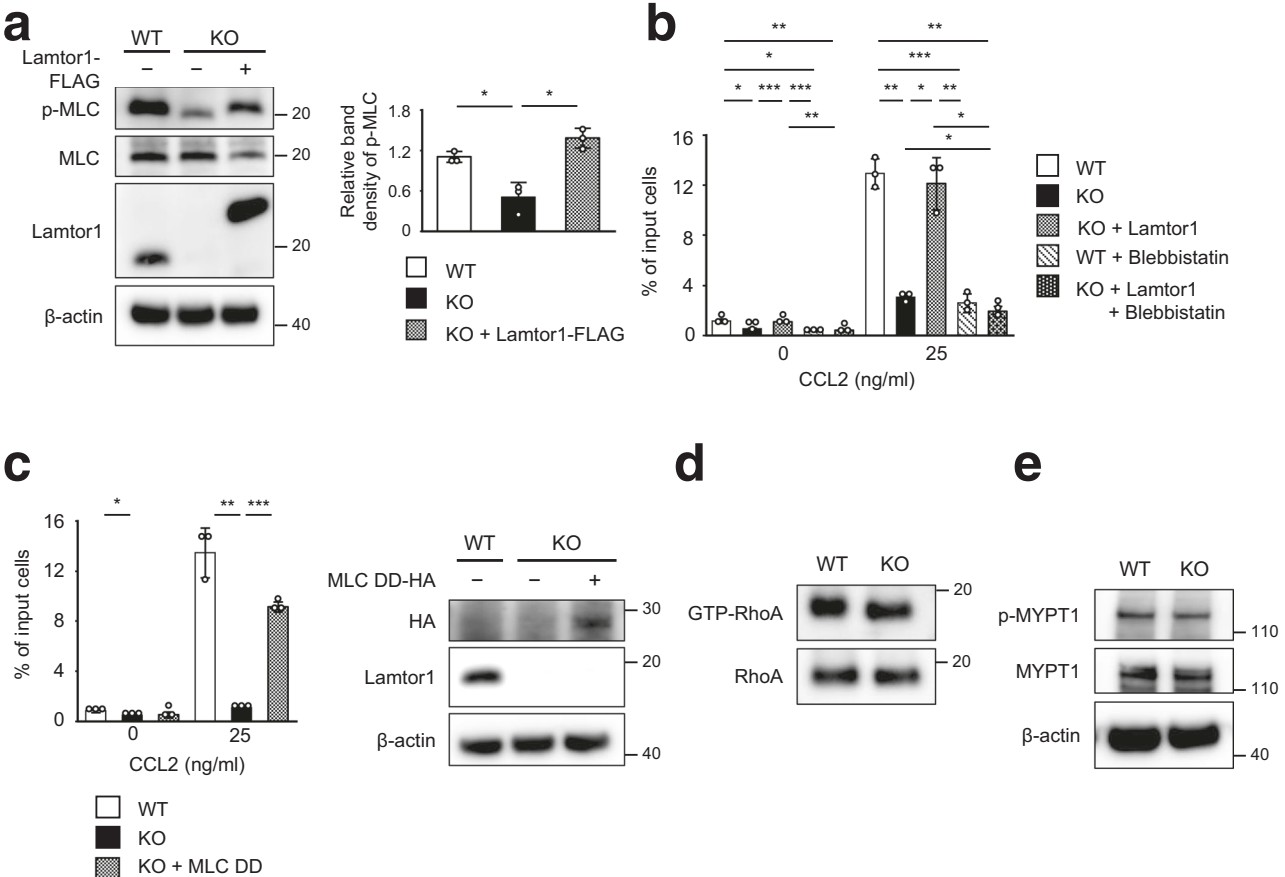

**Fig. 6 Lamtor1-mediated motility is driven by Myosin II activity. a** MLC phosphorylation in the presence or absence of Lamtor1. WT-THP1, *Lamtor1*-KO-THP1, and *Lamtor1*-KO-Full-THP1 cells were lysed, and MLC phosphorylation was detected by western blotting with anti-p-MLC antibody (left). The concentration of p-MLC of WT-THP1 (white bar), *Lamtor1*-KO-THP1 (black bar), and *Lamtor1*-KO-Full-THP1 (mesh bar) in SDS-PAGE gel bands was determined using ImageJ, and statistical analysis was performed (right). **b, c** Myosin II-dependent Lamtor1-mediated motility. Chemotaxis of WT-THP1 (white bar), *Lamtor1*-KO-THP1 (black bar), *Lamtor1*-KO-Full-THP1 (mesh bar), blebbistatin-treated WT-THP1 (diagonal bar), and blebbistatin-treated *Lamtor1*-KO-Full-THP1 (dark mesh bar) in response to 25 ng/ml CCL2 was determined by Transwell assay (pore size, 5 μm) (**b**). Chemotaxis of WT-THP1 (white bar), *Lamtor1*-KO-THP1 (black bar), and *Lamtor1*-KO-THP1 expressing MLC-DD-HA, a constitutively active form of MLC (mesh bar) in response to CCL2 were evaluated by a Transwell assay (pore size, 5 μm) (left). Expression of MLC-DD-HA in *Lamtor1*-KO-THP1 was evaluated by western blotting using an anti-HA antibody (right) (**c**). **d** Active form of RhoA in WT and *Lamtor1*$^{-/-}$ BMDCs. Cells were lysed, and GTP-bound RhoA was pulled down with GST-Rhotekin-RBD and detected by western blotting with an anti-mouse RhoA antibody. **e** MYPT1 phosphorylation in WT and *Lamtor1*$^{-/-}$ DCs. WT and *Lamtor1*$^{-/-}$ BMDCs were lysed and phosphorylated MYPT1 was evaluated by western blotting with an anti-p-MYPT1 antibody. Statistical analysis was performed by two-sided ANOVA with Tukey's post hoc test (**a–c**) [means ± s.d.; *p < 0.05, **p < 0.01, ***p < 0.001]. Data are representative of three experiments.

Moreover, to determine whether adding back Lamtor1 to *Lamtor1*$^{-/-}$ DCs could restore in vivo DC motility, we transduced *Lamtor1*-IRES-EGFP into *Lamtor1*$^{-/-}$ BMDCs using a lentivirus system. Although the efficiency was low, ~7–18% in each experiment, we labeled whole BMDCs exposed to lentivirus with a cell-tracer violet dye and injected them into the footpads of WT mice. We then analyzed the abundance of GFP$^+$ DCs, which represented Lamtor1-restored DCs, in popliteal LNs. The ratio of GFP + DCs to violet+ DCs was higher in popliteal LNs than in the footpad (Fig. 8f), suggesting that adding back Lamtor1 in *Lamtor1*$^{-/-}$ DCs restored in vivo DC motility. Taken together, these results indicate that Lamtor1 is essential for in vivo DC trafficking.

**Impaired in vivo T-cell primings and reduced CHS response in CD11c–*Lamtor1*$^{flox/flox}$ mice.** Next, we examined in vivo antigen-specific T-cell priming following OT-II T-cell transfer and administration of OVA protein with complete Freund's

adjuvant (CFA) into the footpad. T-cell proliferation in dLNs was considerably reduced in CD11c–*Lamtor1*$^{flox/flox}$ mice (Fig. 9a). Because DC maturation is important for T-cell priming, we evaluated the levels of co-stimulatory molecules, including MHC-II, CD80, CD86, and CD40[18]. Expression of these proteins was not impaired in *Lamtor1*$^{-/-}$ BMDCs with or without LPS stimulation, but was slightly higher than in WT BMDCs (Supplementary Fig. 7c), suggesting that DC maturation was not impaired in *Lamtor1*$^{-/-}$ DCs. To assess the in vitro T-cell priming ability of DCs, we performed co-cultures of OVA protein- or OT-II peptide-pulsed DCs and OT-II T-cells. OVA protein- or OT-II peptide-pulsed *Lamtor1*$^{-/-}$ DCs promoted in vitro OT-II T-cell proliferation to the same extent as WT DCs, even though lysosomes are involved in endocytosis and antigen presentation[39] (Supplementary Fig. 7d). These results indicate that impaired DC transport in *Lamtor1*$^{-/-}$ DCs, rather than impaired T–DC interaction or DC activation, caused impaired in vivo antigen-specific T-cell priming. Furthermore, in the hapten-induced contact hypersensitivity (CHS) assay, in which

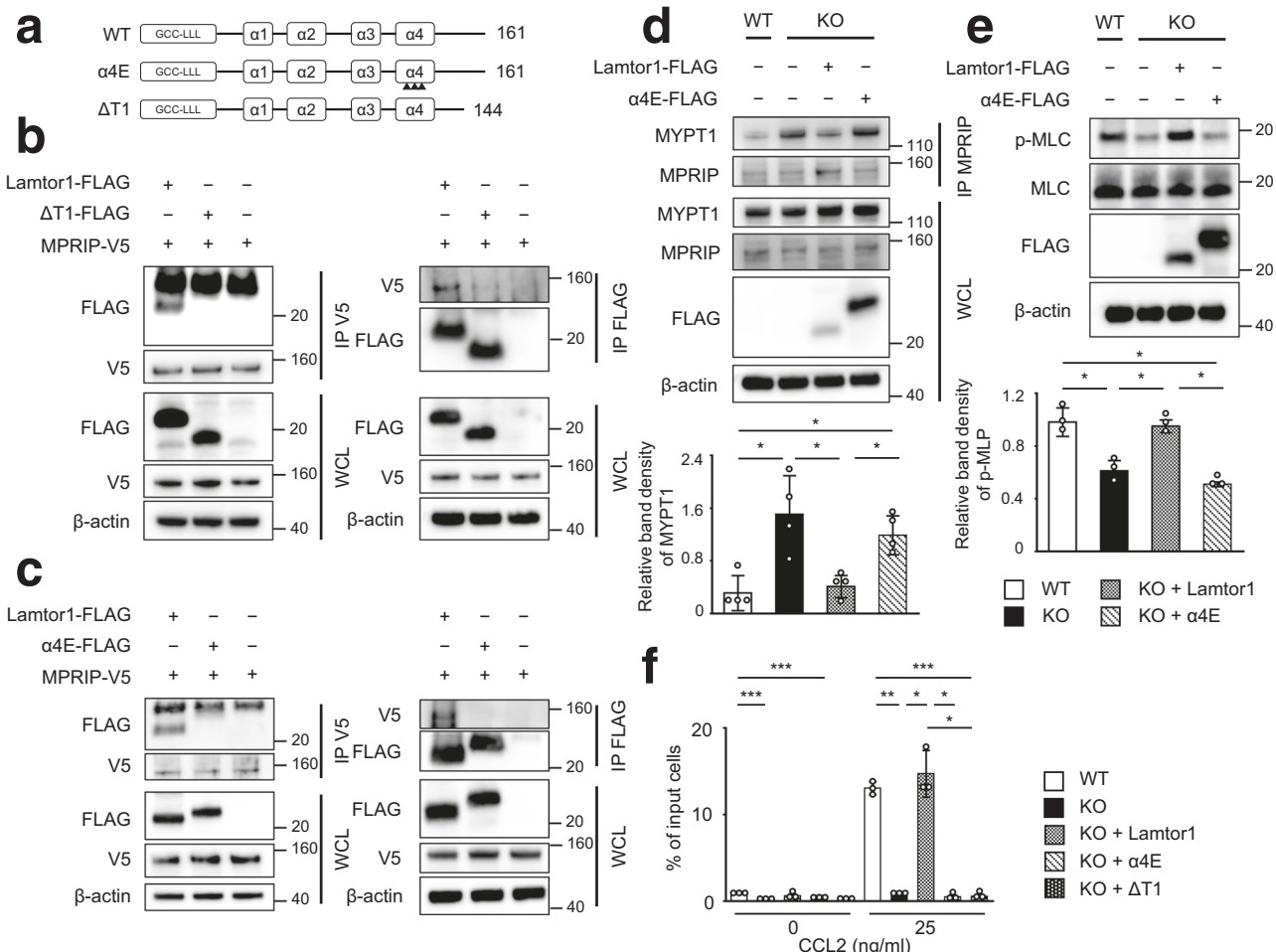

**Fig. 7 The Ragulator complex is essential for interacting to MPRIP, which plays a role in cell migration. a** Schematic of mutant Lamtor1 (α4E and ΔT1). Full-length Lamtor1 (top) and α4E mutant (middle) are Met1–Pro161. Arrowheads indicate the positions of amino acid substitutions (Leu129Glu, Ile135Glu, and Leu143Glu in the α4 helix). ΔT1 is Met1–Ser144 (bottom). **b**, **c** Mutants of Lamtor1 (ΔT1, α4E) failed to interact with MPRIP. FLAG-tagged full-length Lamtor1, a truncated form of Lamtor1 (ΔT1) (**a**), a substituted form of Lamtor1 (α4E) (**b**), or control vector was transfected into V5-MPRIP-expressing HEK293T cells. Cells were lysed, precipitated with anti-V5 (left) or anti-FLAG (right) antibodies, and subjected to western blotting with anti-FLAG and anti-V5 antibodies. **d**, **e** The Lamtor1-α4E mutant did not interfere with the interaction between MPRIP and MYPT1 and failed to restore MLC phosphorylation. Parental THP1 (WT), *Lamtor1*-KO-THP1 (KO), and *Lamtor1*-KO-THP1 cells stably transfected with full-length Lamtor1 or α4E-mutant (Lamtor1-FLAG, α4E-FLAG) were lysed. The cell lysate was precipitated with anti-MPRIP antibody and subjected to western blotting with anti-MYPT1 and anti-FLAG antibodies (upper) (**d**). MLC phosphorylation was detected by western blotting with anti-p-MLC antibody (upper) (**e**). Concentration of MYPT1 co-precipitated with MPRIP (**d**) and p-MLC (**e**) from WT-THP1 (white bar), *Lamtor1*-KO-THP1 (black bar), *Lamtor1*-KO-Full-THP1 (mesh bar), and *Lamtor1*-KO-α4E-THP1 (diagonal bar) in SDS-PAGE gel bands was determined using ImageJ, and statistical analysis was performed (bottom). **f** Chemotaxis of *Lamtor1*-mutant THP1. Chemotaxis of WT-THP1 (white bar), *Lamtor1*-KO-THP1 (black bar), *Lamtor1*-KO-Full-THP1 (mesh bar), *Lamtor1*-KO-α4E-THP1 (diagonal bar), and Lamtor1-KO-ΔT1-THP1 (dark mesh bar) in response to CCL2 (25 ng/ml) was determined by Transwell assay (pore size, 5 μm). Data are representative of three experiments. Statistical analysis was performed by two-sided ANOVA with Tukey's post hoc test (**d–f**) [means ± s.d.; *$p < 0.05$, **$p < 0.01$, ***$p < 0.001$].

mice were sensitized with dinitrofluorobenzene (DNFB) on the abdominal skin and challenged with DNFB on the ear skin[40], ear swelling was lower in CD11c–*Lamtor1*^flox/flox than in WT mice (Fig. 9b).

**Impaired trafficking in *Lamtor1*^−/− neutrophils.** Finally, to investigate the motility of leukocytes other than DCs, we generated LysM-Cre×*Lamtor1*^flox/flox mice, in which Lamtor1 was specifically knocked out in neutrophils and macrophages (Fig. 9c). The fMLP-induced migration of neutrophils was lower in *Lamtor1*^−/− neutrophils isolated from bone marrow than in WT neutrophils (Fig. 9d). In addition, we injected LPS into the peritoneum of WT and LysM–*Lamtor1*^flox/flox mice and counted the number of infiltrated neutrophils in the peritoneal fluid 6 h

after injection by flow cytometry. The percentage of total peritoneal cells that were CD11b⁺Ly6G⁺ and the absolute number of neutrophils were considerably reduced (Fig. 9e). These results indicate that Lamtor1 is indispensable for cell motility not only in DCs, but also in neutrophils.

## Discussion

Leukocytes quickly alter their mode of motility to adapt to environmental demands: mesenchymal motility in the inner vessel walls to confront shear stress, or amoeba-like motility in constricted areas to pass through narrow spaces[19]. In confined 3D environments, they predominantly use the actomyosin contractile force generated by myosin II activity[15, 16]. Phosphorylation of MLC is regulated by MLCP, a heterotrimer consisting of MYPT1,

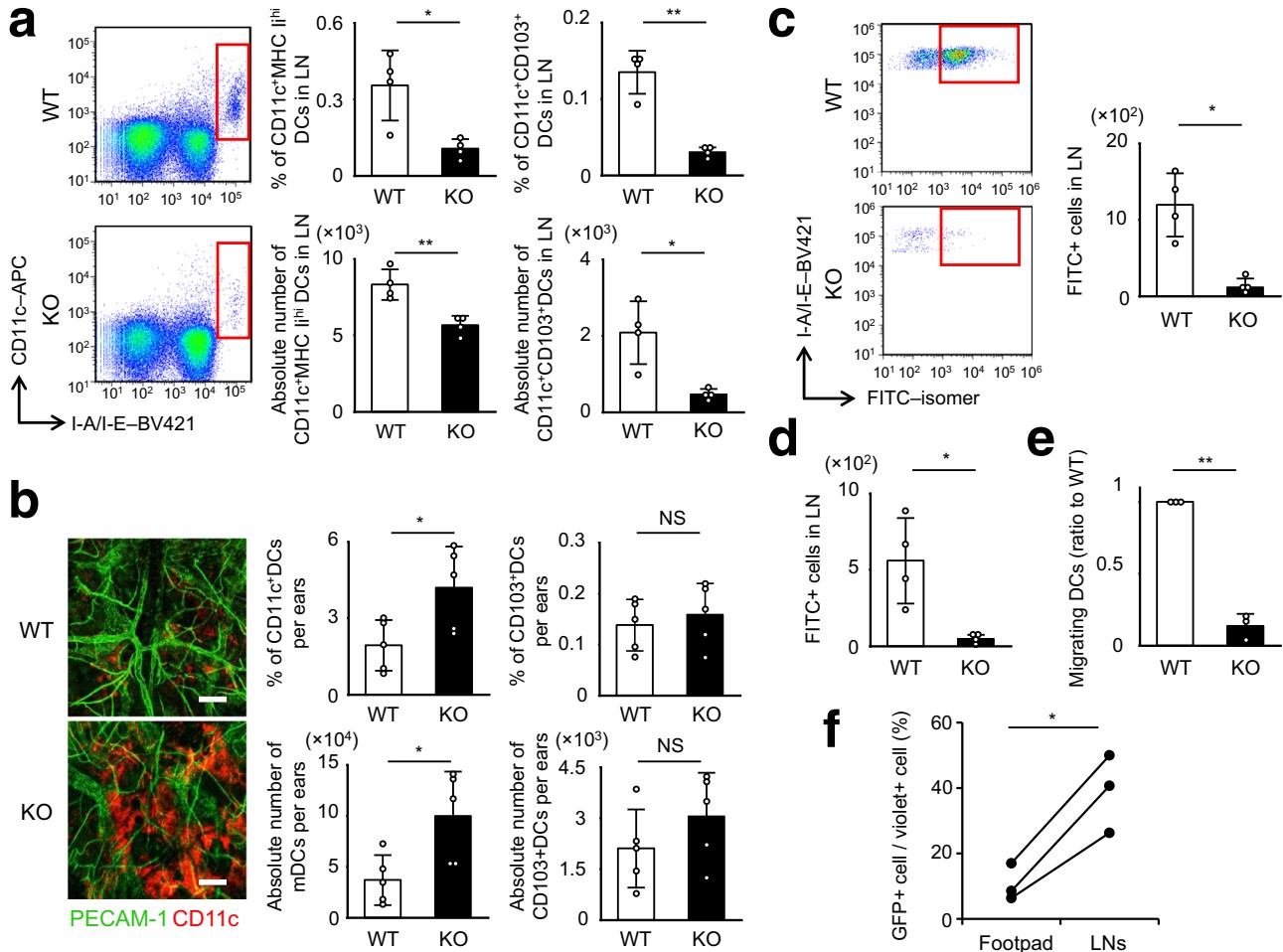

**Fig. 8 Impaired in vivo DC trafficking in DC-specific *Lamtor1*$^{-/-}$ mice. a** Migratory DC subset in inguinal LNs. Single-cell suspension from inguinal LNs was stained for CD11c and MHC II, and migratory DCs (red rectangle) were counted by flow cytometry (left). Population (upper) and absolute number (lower) of CD11c$^+$MHC II$^{hi}$ DCs (middle), and CD11c$^+$CD103$^+$ DCs (right) of WT (white bar) and CD11c–*Lamtor1*$^{flox/flox}$ (black bar) mice are shown. **b** Dermal DCs in the ears. Ears from WT (upper) and CD11c–*Lamtor1*$^{flox/flox}$ (lower) mice were split, fixed in 4% paraformaldehyde, stained with anti-PECAM (green) and anti-CD11c (Red), and observed by confocal microscopy. Scale bar, 100 μm (left). Percentage (upper) and absolute number (lower) of CD45$^+$CD11c$^+$ dermal DCs (middle) and CD11c$^+$CD103$^+$ dermal DCs (right) in WT (white bar) and CD11c–*Lamtor1*$^{flox/flox}$ (black bar) ears, as determined by flow cytometry (right). **c** The number of fluorescein isothiocyanate (FITC)-positive DCs in draining LNs in WT (white bar) and CD11c–*Lamtor1*$^{flox/flox}$ (black bar) mice. Forty-eight hours after FITC-sensitization on the ear, cells isolated from cervical LNs were stained for CD11c and MHC II. FITC$^+$ DCs were evaluated by FACS (red rectangle, left) and counted (right). **d** Number of FITC-bead–bearing DCs in popliteal LNs. Forty-eight hours after FITC-bead injection into the footpads of WT (white bar) and CD11c–*Lamtor1*$^{flox/flox}$ (black bar) mice, cells were isolated from popliteal LNs, stained for CD11c and MHC II, and analyzed by FACS. FITC$^+$ DCs were counted. **e** Comparison of DCs arriving in LNs after footpad administration. CFSE-labeled WT (white bar) and *Lamtor1*$^{-/-}$ (black bar) DCs were adoptively injected into the right and left footpads of WT recipient mice, respectively. Single cells were isolated from popliteal LNs 48 h after injection, and CFSE$^+$ cells were counted. The ratio of the number of fluorescein-labeled DCs was calculated relative to WT. **f** In vivo DC-trafficking of *Lamtor1*$^{-/-}$ DCs in which Lamtor1 expression was restored. *Lamtor1*-IRES-GFP vector was introduced into *Lamtor1*$^{-/-}$ BMDCs using a lentivirus system, and the cells were stained with cell tracer violet. The violet-labeled DCs were injected into the footpad of WT mice, and cells were isolated from popliteal LNs. The ratio of GFP + DCs (Lamtor1-restored DCs) to violet+ DCs (lentivirus-treated DCs including Lamtor1-restored and -not restored DCs) in popliteal LNs was determined by FACS. Data were pooled from four (**a–d**) or three (**e, f**) independent experiments. Statistical analysis was performed by two-sided Student's *t*-test (**a–e**) [means ± s.d.; *$p < 0.05$, **$p < 0.01$] and by two-sided paired sample *t*-test (**f**) [*$p < 0.05$].

M20, and PP1cδ[16]. MLCP activity is regulated by RhoA via the promotion of MYPT1 phosphorylation, which inactivates the catalytic activity of PP1cδ[22]. However, MYPT1 phosphorylation and GTP-RhoA levels were comparable, regardless of the presence or absence of Lamtor1. Peripherally transported lysosomes have been proposed to be important to leukocyte migration. That is, the release of proteases by lysosomal exocytosis mediated by synaptotagmin allows the uropod to be detached from the ECM[14]; mucolipin-1, a lysosomal transient receptor potential cation channel, regulates spatiotemporal cytoskeletal dynamics[13]. In this study, we revealed the significance of peripherally

transported lysosomes in motile cells, in which the Ragulator complex inactivates MLCP by interacting with MPRIP. This interaction interferes with the binding between MPRIP and MYPT1 and ultimately facilitates myosin II activity. That is, in immotile cells, the Ragulator complex localized to lysosomes distributed preferentially in the perinuclear region, and MPRIP anchors MLCP on myosin–actin bundles by binding MYPT1, a subunit of MLCP, resulting in suppression of MLC phosphorylation. In motile cells, lysosomes bearing the Ragulator complex move to the uropod, where the Ragulator complex interacts with MPRIP. This interaction interferes with the interaction between

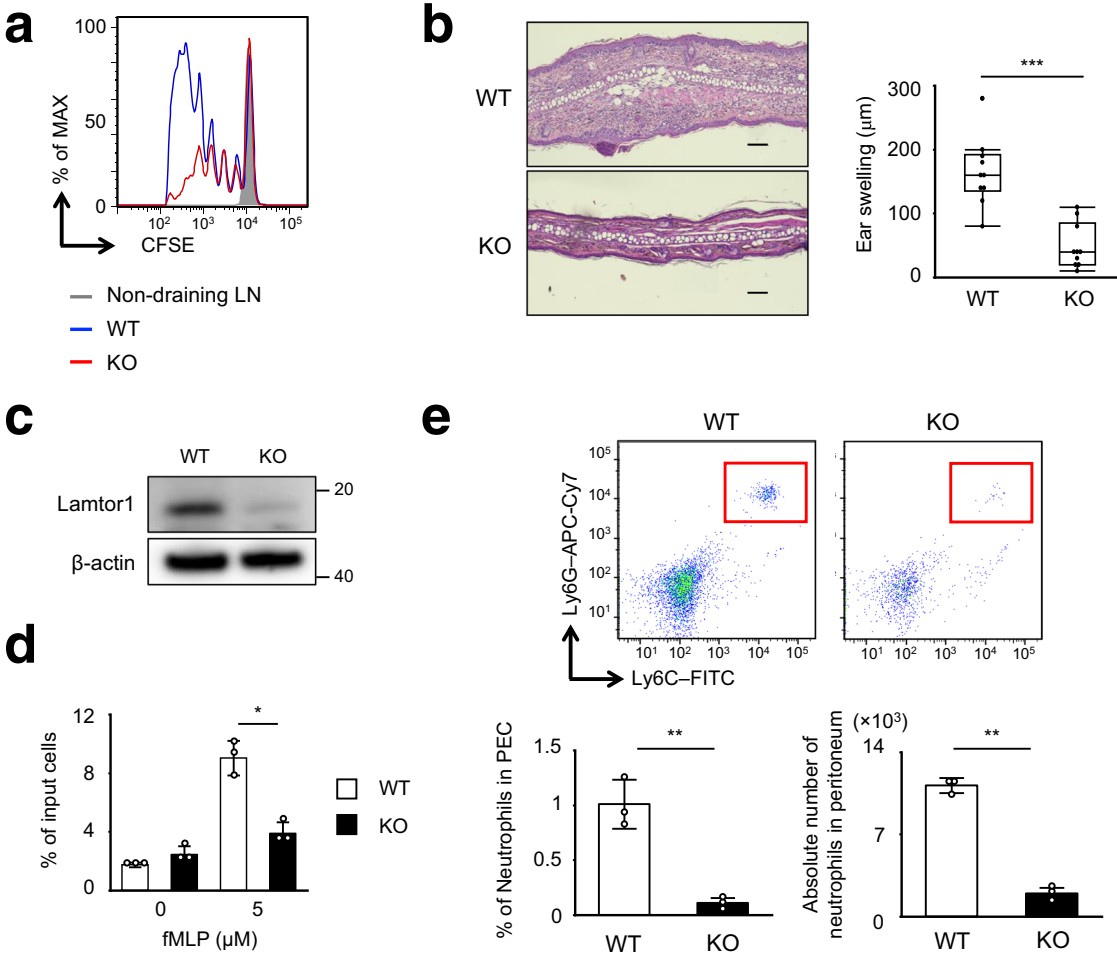

**Fig. 9 Impaired in vivo T cell priming and neutrophil trafficking under Lamtor1 deficiency. a** In vivo OT-II T-cell proliferation following subcutaneous injection of OVA protein. After intravenous transfer of CFSE-labeled CD4+ OT-II T cells into WT (blue line) or CD11c–Lamtor1flox/flox (red line) recipient mice, OVA protein in complete Freund's adjuvant was subcutaneously injected into the footpads of recipient mice. CFSE-dilution of OT-II T cells in popliteal LNs was analyzed by FACS. Data are representative of three independent experiments. **b** Hapten-induced contact hypersensitivity (CHS) in WT and CD11c–Lamtor1flox/flox mice. Five days after sensitization with DNFB in acetone, DNFB was applied to the ears. Forty-eight hours after the DNFB challenge, histological analysis (left) and evaluation of ear thickness (right) were performed (n = 10). Representative images are shown of WT (top) and CD11c–Lamtor1flox/flox (bottom) mice. Scale bar, 100 μm. **c** Lamtor1 expression in neutrophil of WT and LysM-Lamtor1flox/flox mice. Neutrophils isolated from BM from WT and LysM-Lamtor1flox/flox mice (KO) were lysed, and Lamtor1 expression was evaluated by western blotting. Data are representative of three experiments. **d** Chemotaxis of neutrophils in response to fMLP. Chemotaxis of WT (white bar) and Lamtor1−/− neutrophils (black bar) in response to fMLP (5 μM) was determined by Transwell assay (pore size, 3 μm). Data are representative of three experiments. **e** LPS-induced in vivo neutrophil mobilization. LPS was administered into the peritoneum of WT (WT, white bar) and LysM-Lamtor1flox/flox mice (KO, black bar). Six hours after administration, cells in the peritoneal cavity were isolated and stained for CD11b, Ly6C, and Ly6G, and the number of CD11b+Ly6G+ neutrophils (red rectangle) was counted by FACS (upper). Population (left) and absolute number (right) of CD11b+Ly6G+ neutrophils are shown (lower) (n = 3). Statistical analysis was performed by two-sided Student's t-test (**d**, **e**) [means ± s.d.; *p < 0.05, **p < 0.01] or two-sided Mann–Whitney U-test (**b**) [median, 25th and 75th percentiles, and minimum and a maximum of a population excluding outliers; ***p < 0.001].

MPRIP and MYPT1 and decreases MLCP activity, thereby increasing MLC phosphorylation. Consequently, cell motility is facilitated (Fig. 10). This represents a mechanism for tuning the phosphorylation level of MLC.

We showed that phenotypes of Lamtor1-deficiency could not be restored by adding back Lamtor1 mutants that failed to form the Ragulator complex. In the absence of Lamtor2/3, all other components of the Ragulator complex are rapidly degraded via the ubiquitin–proteasome pathway[41]. We also showed that expression of the Ragulator complex's component proteins was lower in Lamtor1-KO cells and Lamtor1-mutants restored cells, suggesting that the constituent molecules of the Ragulator complex cannot exist stably and separately and that the Ragulator complex can exist in leukocytes only when it forms with the proper structure. Consistent with our findings, DC-specific Lamtor2−/− mice have smaller numbers of migratory DCs in skin-dLNs, and all components of the Ragulator complex were less abundant in Lamtor2−/− DCs[42].

We showed that treatment with rapamycin at levels that effectively suppress S6K phosphorylation does not inhibit DC migration, MLC phosphorylation, or the Lamtor1–MPRIP interaction, although mTORC1 is involved in Lamtor1-mediated macrophage polarization[8]. Because DCs are so fast-moving (DCs, 5–20 μm/min vs. mesenchymal cells, 0.1–0.5 μm/min)[19], the interaction between the Ragulator complex and MPRIP must involve spatiotemporal regulation of the cytoskeleton rather than the activation of transcription and protein synthesis through mTORC1.

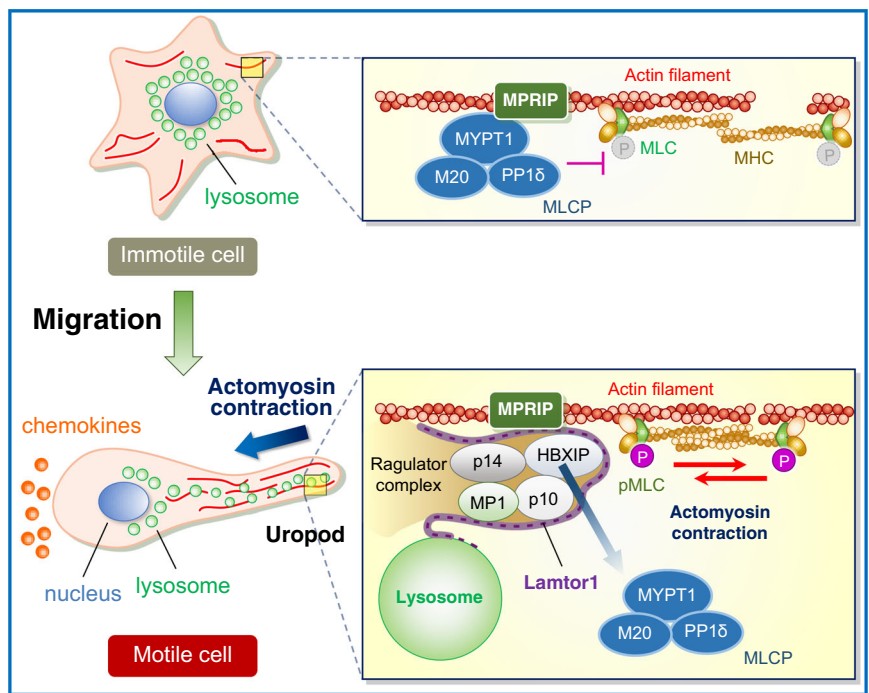

**Fig. 10 Schematic of the mechanism of regulation of myosin II activity by the Ragulator complex.** In immotile cells (upper), the Ragulator complex localized to lysosomes is preferentially distributed in the perinuclear region (left), and MPRIP anchors MLCP on myosin–actin bundles by binding MYPT1, a subunit of MLCP, resulting in suppression of MLC phosphorylation (right). In motile cells exposed to chemokines (lower), the lysosomes bearing the Ragulator complex moves to the uropod (left), where the Ragulator complex interacts with MPRIP. This interaction interferes with the interaction between MPRIP and MYPT1 and decreases MLCP activity, thereby increasing MLC phosphorylation. Consequently, cell motility is facilitated (right).

Mutation in the 3′ region of Lamtor2 causes a human immunodeficiency syndrome in which patients exhibit a delay in bacterial degradation and an impaired memory response[2]. Our observations showing that the Ragulator complex is indispensable not only for DCs but also for neutrophil trafficking may provide an insight into the causes of human immunodeficiency syndrome. Also, the interaction between the Ragulator complex and MPRIP represents a therapeutic target for autoimmune diseases and inflammation by inhibiting leukocyte motility. Future studies should seek to identify and characterize the Ragulator–MPRIP interaction site, as well as compounds that inhibit the interaction.

## Methods

**Mice.** *Lamtor1*flox mice were generated by Dr. Shigeyuki Nada as described elsewhere[43]. CD11c-Cre transgenic mice were provided by Dr. Hiroaki Henmi. LysM-Cre transgenic mice were provided by Dr. Shizuo Akira. Genotyping was performed using primers 5′-AAGGATTCGGAGTTAGAGACTAGGAC-3′ and 5′-TGAGGATTCGAGTGGTGAGATACGA-3′ for *Lamtor1*flox and *Lamtor1* alleles, 5′-AGACTCAGCTCAAGTGCTAC-3′ and 5′-GCGAACATCTTCAGGT TCTG-3′ for CD11c-Cre-Tg mice, and 5′-CTTGCTGTGTGTTGTTCTGTGCT GAGG-3′ and 5′- GCATAACCAGTGAAACAGCATTGC-3′ for LysM-Cre-Tg mice. DC-specific *Lamtor1*−/− mice and myeloid cell-specific *Lamtor1*−/− mice were generated by crossing the two parental strains and backcrossing the progeny ~10 times to C57BL/6J. Mice were housed under specific pathogen-free conditions with a 12 h light/dark cycle, at a temperature of $22 ± 2$ °C, and relative humidity of $50 ± 5\%$. Mice were fed a standard mouse chow diet and 3 to up to 5 mice were housed in the same cage. Eight-to-twelve-week-old mice were used for experiments. Both male and female mice were used; however, sex was always matched in each experiment. The application of animal experiments was approved by the ethical board of the graduate school of medicine, Osaka University (28-008-034), and all animal experiments were performed according to Osaka University's regulations.

**Cells and cell culture.** BMDCs were generated by culturing bone marrow cells with 50 μg/ml GM-CSF (R&D Systems) for 6–8 days as described[44]. THP1, a human acute monocyte leukemia cell line, was obtained from ATCC (TIB-202) and cultured in RPMI 1640 supplemented with 10% FCS, 500 μg/ml penicillin, and streptomycin, and 0.05 mM 2-mercaptoethanol. HEK293T cell line was obtained from ATCC (CRL-3216) and cultured in DMEM supplemented with 10% FCS and 500 μg/ml penicillin and streptomycin.

**Antibodies, reagents, and fluorescent dyes.** Reagents were obtained from the indicated suppliers: LPS, PMA, rapamycin, FITC-isomer I, fibronectin OVA (Grade VI) CFA (Sigma); 2,4-DNFB (Tokyo Chemical Industry); collagenase D (Roche); type I collagen (BD Biosciences); calcein-AM, CFSE (carboxyfluorescein diacetate succinimidyl ester), phalloidin-546 (Invitrogen); AcidiFluor-ORANGE (GORYO Chemical); Torin1 (Selleck); U0216 (Calbiochem); blebbistatin (Cayman Chemical); recombinant mouse CCL19, mouse CCL21, human MCP-1 (R & D Systems); mouse GM-CSF (Wako); and Evans blue (Nacalai Tesque). All of the antibodies we used were listed in Supplementary Table 2. These commercially available antibodies were checked the validation data on antibody quality, including specificity, application, and immune species in the vendor's website.

**Flow cytometry.** For screening of immune cell populations and expression of co-stimulatory molecules in DCs, a single-cell suspension isolated from spleen or skin-dLNs from 6 to 8-week-old mice, BMDCs, or LPS-induced peritoneal infiltrating cells were stained with the indicated antibodies, and data was collected by a FACS Canto II (BD Biosciences) by FACS Diva (v5.0.3) software and analyzed by Flow Jo (v10) software.

**In vivo DC trafficking assay.** For the FITC-skin sensitization assay, 50 μl of 2 mg/ ml FITC isomer I in carrier solution (1:1 vol/vol, acetone: dibutyl phthalate) was administered to the ears. Control animals received the same amounts of carrier solution without FITC[38]. The mice were sacrificed after 48 h, and the cervical LNs (dLNs of the ear) was removed and treated with collagenase D to obtain a single-cell suspension. For the FITC-microbead injection assay, mice were subcutaneously injected with 1-μm diameter FITC beads in their footpads. After 48 h, the popliteal LNs were removed and treated with collagenase D to obtain a single-cell suspension. For the fluorescein-labeled DC-injection assay, $3 × 10^6$ BMDCs labeled with 5 μM CFSE were injected into the footpads of WT mice. After 48 h, the popliteal LNs were removed and treated with collagenase D to obtain a single-cell suspension. The frequency of FITC- or CFSE-positive CD11c+MHC-II+ DCs was assessed by flow cytometry.

**In vivo immune response assay.** Ag-specific T-cell proliferation assay was performed as described elsewhere[25]. Briefly, $10^7$ cells/ml of CD4+ OT-II T cells were labeled with 1 μM CFSE in PBS (pH 7.4) for 10 min, and $5×10^6$ cells were intravenously transferred into each recipient mouse. After 2 h, the mice were immunized subcutaneously with OVA in CFA in the hind footpads (Grade VI, Sigma). After 72 h, mice were sacrificed, the popliteal LNs were removed, and Ag-specific T-cell proliferative responses were analyzed by measuring the dilution of CFSE-fluorescence. CHS was induced as previously described with modifications[45].

Briefly, 30 μl of 0.5% (vol/vol) 2,4-DNFB dissolved in acetone/olive oil (4:1) was administered to the shaved abdominal skin. After 5 days, the side of the ear was challenged with 0.2% (vol/vol) DNFB (10 μl into the ear). Ear thickness was measured with a micrometer (Mitsutoyo, Kawasaki, Japan) 48 h after DNFB treatment.

**In vitro cell migration assay**. For the Transwell assay, inserts (pore size, 5.0 μm for DCs and THP1, 3.0 μm for neutrophils; Corning) were placed in 24-well plates filled with 0.6 ml of 0.1% (wt/vol) BSA in RPMI medium containing recombinant CCL19 or CCL21 (for DCs), CCL2 (for THP1), and fMLP (for neutrophils). A cell suspension of DCs, THP1, or neutrophils ($2 \times 10^5$ cells/100 μl) was added to the upper well, followed by incubation for 4 h at 37 °C. Cells in the lower chamber were detached for 5 min with 5 mM PBS-EDTA and counted on a Guava PCA system. Migration ability (% of input cells) was evaluated by dividing the number of cells in the lower chamber by the number of cells input into the upper chamber and multiplying by 100. For DC migration in 3D collagen matrices, chemotaxis assays were performed using a Zigmond chamber as described elsewhere[25]. Briefly, lipopolysaccharide (200 ng/ml)-treated BMDCs were suspended in 2 mg/ml type I collagen (BD Biosciences) containing 5% FCS and placed on one side of the Zigmond chamber to cover the stage with gel. Cells were incubated for 30 min at 37 °C to allow polymerization of the matrix, and then RPMI medium containing 0.1% BSA with CCL19 (5 μg/ml) was added to the other chamber. After 20 min of incubation, DC locomotion was examined at 1-min intervals by confocal time-lapse video microscopy. The velocity of cells was measured using the ImageJ manual tracking software. To visualize lysosomes, BMDCs were labeled with 1 μM AcidiFluor-ORANGE for 2 h before being subjected to chemotaxis assays in a Zigmond chamber.

**Western blotting**. Two million BMDCs or BM-derived neutrophils, $2 \times 10^6$ parental or stably transfected THP1 cells, or $1 \times 10^6$ HEK293T cells were solubilized in buffer A [1% Nonidet P-40 (NP-40), 50 mM Tris–HCl (pH 7.4), 150 mM NaCl, 1 mM EDTA, 5% glycerol, 2% n-octyl-b-D-glucopyranoside] with proteinase inhibitor (Roche) and centrifuged at 15,000 rpm for 20 min at 4 °C. Supernatants were mixed with 2× Laemmli sample buffer containing 2-mercaptoethanol and denatured at 95 °C for 3 min. The reduced samples were electrophoresed in 4–12% Bis–Tris gels (Life Technologies), transferred to nitrocellulose membranes and blotted with the antibodies listed in Supplementary Table 2. Protein concentrations in SDS–PAGE gel bands were determined using the ImageJ software (v1.53a), and statistical analysis was performed using the two-sided Student's *t*-test.

**Immunoprecipitation**. V5-tagged MPRIP-expressing HEK293T cells were transfected with FLAG-tagged wild-type or mutant Lamtor1 using FuGENE6 (Promega) and incubated for 48 h. Cells were solubilized with buffer A and centrifuged at 15,000 rpm for 10 min at 4 °C. Immunoprecipitation was performed using the Dynabeads Protein G Immunoprecipitation Kit (10007D, Invitrogen). Briefly, after binding of the antibody to magnetic beads (10 min at room temperature, with rotation), cell lysates and antibody-coated beads were mixed and incubated for 30 min at room temperature. After three washes, proteins were eluted with elution buffer. Immunoprecipitates were mixed with 2× Laemmli sample buffer containing 2-mercaptoethanol and denatured at 95 °C for 5 min. The samples were separated by SDS–PAGE and blotted with anti-V5 or anti-FLAG antibodies. For analysis of the interaction between MPRIP and MYPT1, parental THP1, Lamtor1-KO-THP1 and stable transfectants (Lamtor1-KO-Full-THP1 and Lamtor1-KO-α4E-THP1) were solubilized with buffer A, pre-cleared with protein G–agarose, and immunoprecipitated with protein G–agarose plus anti-MPRIP Ab (Cell Signaling Technologies) overnight at 4 °C. After five washes with lysis buffer A, the immunoprecipitates were separated by SDS–PAGE and blotted with anti-MYPT1 antibody (Cell Signaling Technologies).

**Immunohistochemistry**. For in vitro immunohistochemical analysis, cells were plated on fibronectin (10 μg/ml)-coated glass-bottom dishes (Iwaki). For DCs, cells were stimulated with CCL19 for 30 min and fixed in 4% paraformaldehyde (PFA) for 20 min. For THP1, cells were activated with PMA (50 nM) for 2 days and then fixed in 4% PFA for 20 min. Cells were classified as polarized or non-polarized based on morphology. For histological analysis of the dermis of the ear, mouse ears were split into front and back, and then fixed in 4% paraformaldehyde for 20 min. After washes with PBS containing Triton X-100 (0.1% vol/vol), the samples were blocked with BSA (5% wt/vol) in PBS for 60 min, and then stained with primary antibodies at 4 °C overnight. After four washes with PBS containing 0.1% Triton X-100, the samples were stained with fluorescently labeled secondary antibodies for 1 h. The samples were mounted in PermaFluor mounting medium (Thermo Scientific). All fluorescence images were obtained by confocal microscopy (LSM 710, Zeiss). To evaluate co-localization, a Z-stack image was acquired by optimizing the pinhole size according to the objective lens (×63). Representative slice images are shown in the corresponding figures.

**Expression vector cloning**. For protein expression, cDNA encoding Lamtor1 (Met1-Pro161) fused C-terminally to the 3×FLAG tag was cloned into the lentiviral vector CSII-EF-MCS-IRES2-Venus (provided by Dr. Hiroyuki Miyoshi, Keio

University). The lentiviral vector encoding V5-tagged MPRIP (Met1-Asp2457)-IRES-RFP was produced by VectorBuilder. A truncated cDNA fragment of human *Lamtor1* (ΔT1, Met1-Ser144, deletion of C-terminal tail; ΔT2, Met1-Gln114, deletion of α3, α4 helix and C-terminal tail; ΔT3, Met1-His41, deletion from α1 to C-terminal tail) was ligated to the 3×FLAG sequence by overlapping PCR using the primers listed in Supplementary Table 4, and cloned into the *Not*I-*Bam*HI sites of the CSII-EF-IRES2-Venus vector using the In-Fusion HD Cloning Kit (TaKaRa). Mutant cDNA of α4E-Lamtor1 (Leu129Glu, Ile135Glu, and Leu143Glu in the α4 helix) was ligated to the 3×FLAG sequence by overlapping PCR using the pETDuet-α4E-Lamtor1 plasmid as a template, and cloned by the method described above. Mutant cDNA of DD-MLC (replacement of Thr18Asp and Ser19Asp) fused to the HA sequence was synthesized by Eurofin and cloned into the *Not*I-*Hpa*I sites of vector CSII-EF-IRES2-Venus using the In-Fusion HD Cloning kit.

**Generation of Lamtor1-deficient THP1 cell lines using CRISPR-Cas9**. THP1 cells stably expressing Cas9 (THP1 Cas9) were generated as previously described[46]. In order to generate the Lamtor1-deficient cell line, the cells were transfected with a mixture of several gRNAs targeting *Lamtor1* using Avalanche-Omni transfection reagent (EZ Biosystems). Target sequences used for the *Lamtor1* knockout were 5′-TGCGAGCGGAAGGCAGGCTG-3′, 5′-GCTCCGGGACAG GGGGTACG-3′, 5′-TCCGGTCACATGACCCGCGG-3′, 5′-CTGCTACAGCAG CGAGAACG-3′, 5′-GGCCTGCTCATCAGTGCGAG-3′, and 5′-AGGTGCTCA CCTGTACTGCC-3′. Briefly, gRNA was synthesized using the CUGA7 gRNA Synthesis Kit (Nippon Gene). A mixture of 20 ng/sgRNA and 0.14 μl Avalanche-Omni was added to the culture medium of THP1-Cas9 in a 96-well plate ($5 \times 10^4$ cells/well). After isolation of monoclonal cell populations, knockout of Lamtor1 was confirmed by western blotting.

**Establishment of stable transfectants**. *Lamtor1*$^{-/-}$ THP1 transfectants stably expressing full-length or mutant Lamtor1, or MLC-DD, were established by lentivirally introducing FLAG-tagged full length-Lamtor1 (Lamtor1-KO-Full-THP1), FLAG-tagged ΔT1-Lamtor1 (Lamtor1-KO-ΔT1-THP1), FLAG-tagged α4E-mutant-Lamtor1 (Lamtor1-KO-α4E-THP1), or HA-tagged MLC-DD mutant (Lamtor1-KO-MLC-DD-THP1) into *Lamtor1*$^{-/-}$ THP1 as previously described[46]. Transfectants were selected in the presence of puromycin, and single clones were screened by western blotting with anti-FLAG mAb (M2, Sigma) or anti-HA mAb (Y-11, Santa Cruz Biotechnology).

**Gene silencing with shRNA**. Expression of MPRIP in THP1 was silenced by the lentiviral introduction of an shRNA targeting MPRIP (TRCN0000048648) or a non-target control (Sigma) and selected in the presence of 1.0 μg/ml puromycin (Invivogen). Knockdown of MPRIP was confirmed by western blotting.

**RhoA pull-down assay**. After BMDC lysis, the active form of RhoA was pulled down with GST-Rhotekin-RBD fusion protein and immunoprecipitated with glutathione resin (Active RhoA Detection Kit, Cell Signaling Technologies). Active RhoA levels were determined by western blotting using anti-mouse RhoA antibody (Cell Signaling Technologies).

**Adhesion assay**. A half-million calcein-AM (5 mM)-stained BMDCs in RPMI supplemented with BSA (0.1% wt/vol) were applied to fibronectin (1, 2, 5 mg/ml)-coated wells. After incubation at 37 °C for 30 or 90 min, non-adherent cells were completely removed by washing with PBS, and attached cells were denatured in 0.1% NP40. The optical density of each well was determined using a microplate reader set to 485 nm.

**In vitro DC migration assay in a 2D environment**. For the random cell migration assay, 5000 BMDCs in RPMI1640 containing 0.1% (wt/vol) BSA were applied to fibronectin (10 mg/ml)-coated μ-Dish 35 mm Quad (Ibidi). Phase-contrast images of migrating cells were acquired at 1-min intervals. For horizontal directional migration assays, chemotaxis assays were conducted in an EZ-TAXIScan chamber (Effector Cell Institute). Briefly, one compartment contained CCL19 (5 mg/ml), and the other contained BMDCs; the two compartments were connected by a microchannel. Phase-contrast images of migrating cells were acquired at 30-s intervals.

**Anatomical analysis of the lymphatics**. Evans blue dye (5 mg/ml) was injected into the footpad; after 3 h, the mice were sacrificed, and afferent lymphatics and popliteal LNs were visualized based on Evans blue staining.

**LC-MS/MS analysis**. After BMDCs were lysed in lysis buffer A (50 mM Tris–HCl (pH 7.4), 150 mM NaCl, 1 mM EDTA, 5% glycerol, 2% n-octyl-b-D-glucopyrano-side, and 1% NP-40), immunoprecipitation was performed using an anti-Lamtor1 antibody. LC-MS/MS of immunoprecipitates was performed on Q-Exactive and UltiMate 3000 Nano LC systems (Thermo Fisher Scientific, Yokohama, Japan). Data

were analyzed using the Scaffold software (version 4.4.3) to obtain a list of protein identifications with a false discovery rate of 0.5%.

**Statistical analysis**. All statistical analyses were conducted using JMP Pro version 12.2.0. Normally distributed data were compared using the two-sided Student's $t$-test, ANOVA with Tukey's post hoc test, or paired sample $t$-test, and are presented as means ± SD. Non-normally distributed data were analyzed using the Mann-Whitney $U$ test, and are expressed as medians, 25th and 75th percentiles, maximum, and minimum of the population, excluding outliers if necessary. A value of $P < 0.05$ was considered significant.

**Reporting summary**. Further information on research design is available in the Nature Research Reporting Summary linked to this article.

## Data availability

LC/MS data was available from JPOST (WT, JPST001140, PXD025612; Lamtor1-KO, JPST001141, PXD025611). All other data that support the conclusions are available from the corresponding author on reasonable request. Source data are provided with this paper.

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

## Acknowledgements

We are grateful to Dr. Hiroyuki Miyoshi (Keio University, Japan) and the RIKEN BioResource Center (RIKEN BRC) for providing materials. We also thank Dr. Hiroaki Henmi (Wakayama Medical University, Japan) for providing CD11c-Cre-Tg mice, and Shizuo Akira (Osaka University, Japan) for providing LysM-Cre-Tg mice. This study was supported in part by a Core Research for Evolutionary Science and Technology (CREST) grant from the Japan Science and Technology Agency (JST) (JPMJCR16G2, to H.T.); the Center of Innovation program (COI-STREAM) from the Ministry of Education, Culture, Sports, Science and Technology of Japan (MEXT) (to A.K.); the Japan Agency for

Medical Research and Development (AMED)-CREST (15652237, to A.K.); the Japan Society for the Promotion of Science (JSPS) KAKENHI (JP18H05282 to A.K.); and the Japan Agency for Medical Research and Development (AMED) (JP18cm016335 and JP18cm059042 to A.K.). The authors have no conflicting financial interests.

## Author contributions

H.T. and T.N. designed the project. T.N. performed most of the experiments and analyzed the data. K.T., J.P., and T.J. generated the vectors used in this work and established the mutant cell lines. S.N. and M.O. generated *Lamtor1*$^{-/-}$ mice, provided the mutant Lamtor1 vectors, and made critical collaborative suggestions. H.T. and T.N. wrote the manuscript. Y.H. and H.K. assisted with the experiments. T.K., T.M., Y.K., M.N., and S. K. participated in discussions. A.K. supervised the study.

## Competing interests

The authors declare no competing interests.
