## [Peer Review File · Nature Communications]

REVIEWER COMMENTS

Reviewer #1 (Remarks to the Author):

Nakatani et al. present a comprehensive, engaging research paper that is of interest to the immunology community, bridges a knowledge gap, and highlights potential clinical applications.

Broadly, the group investigated the role of the lysosome-bound Ragulator complex in the motility of dendritic cells. They show that Lamtor-1, one of the components of the pentameric Ragulator, colocalizes with lysosomes in the uropod of polarised bone marrow derived dendritic cells (BMDCs). Lamtor-1 also colocalized with phosphorylated myosin light-chain (p-MLC). Both MLC phosphorylation and BMDC migration were impaired in Lamtor-1^{-/-} BMDCs.

The group further showed that Lamtor-1, when it is able to form the full Ragulator complex, coprecipitates and interacts with MPRIP. MPRIP was previously shown to anchor the MYPT1 subunit of MLC phosphatase to the actin-myosin filament, effectively reducing MLC phosphorylation and thus contractility. Nakatani et al. propose that the Ragulator complex binds to MPRIP and interferes with the MPRIP-MYPT1 interaction, thus blocking the recruitment of MLC phosphatase and resulting in more p-MLC and increased contractility. They also confirm that Lamtor-1 is required for DC migration *in vivo*.

Interestingly, despite Ragulator acting as a scaffold for mTORC1, the addition of mTORC inhibitors had no effect on DC motility or the interaction between Lamtor-1 and MPRIP, and S6K phosphorylation was not affected in the Lamtor-1 knockout.

The paper is well-written and presents a logical story. However, there are several points that require clarification or elaboration.

Firstly, to strengthen the title and the conclusions of this paper, I suggest that the main experiment be repeated with macrophages or other immune cells. The authors refer to Lamtor's relevance in macrophages in the introduction (line 62), but then focus on DCs without a clear justification – monocytes, macrophages, T-cells and neutrophils have all been shown to move at the same velocity as DCs. DCs use different migration strategies than other immune cells (Nature Reviews Molecular Cell Biology volume 20, pages738–752 (2019). Repeating, for instance, the immunofluorescence study with macrophages could either broaden the scope of the findings or highlight a new DC-specific mechanism depending on the results. This would also justify the authors' claim in line 235 that Ragulator is "indispensable for leukocyte trafficking". The conclusion also states that the newly discovered Ragulator/MPRIP interaction may provide a target for combating cancer metastasis, which is not mentioned or explained anywhere else in the text – the authors should justify this claim.

Although the authors did not find a link between their results and mTORC1 activity, they themselves state that the findings on mTORC1's role in leukocyte activity have thus far been controversial. The authors may wish to consider including information on the amino acid content of their chosen medium (RPMI), and could repeat the mTORC1 inhibition experiments under more nutrient-rich conditions (e.g. in DMEM) or minimal conditions.

The authors present a large amount of western blot data. However, the methods and figure legends do not sufficiently describe the detection methods that were employed. These should be added for reproducibility. Additionally, none of the blots were quantified. As western blotting is often used to

show relative amounts, quantification is often not required. However, to strengthen certain claims quantification would be useful; for example for Figures 2G and 3C.

There is also a lack of clarity regarding the polarized state of DCs. The authors generally use CCL19, but include CCL21 in Figure 1D. DC motility may vary in response to different chemokines (see e.g. Schumann et al. (2010) in *Immunity*), so it may be interesting to repeat some experiments with CCL21 as the polarizing agent. Regardless, if “polarized” always specifically refers to “CCL19-treated”, this should be clarified. It is unclear whether the cells in Figure 1J were treated in some way to induce polarization.

On another small note, there are some layout issues. For example, the figure legend for Figure 1D refers to the “left” and “right” graphs, when the two graphs are in fact stacked vertically. Additionally, the placement of Supplementary Figure 1B is odd in terms of the paper’s chronology, as it is first referred to very late in the text, only after Supplementary Figure 4. Supplementary Figure 1A shows the merging of the two stains shown in Figures 1B and 1C, and since the authors refer to colocalization, they may want to consider including the merged image in the main text, as this is more convincing. Finally, I would consider rephrasing that DCs are “professional cells” (line 175); perhaps changing it to the implied “professional antigen-presenting cells”.

There is a small lack of information regarding the FACS assay of dendritic cells, which are classified as “migratory” based only on their expression of MHC-II. There are multiple diverse subsets of DCs with a variety of different markers. For more accuracy, CD103 should be included. The authors may want to consider including or explaining their gating strategy as well.

Reviewer #2 (Remarks to the Author):

In the manuscript “The lysosomal Ragulator complex plays an essential role in 2 leukocyte trafficking by activating myosin II “ by Nakatani et al, the authors describe new mechanistic understanding of how lysosomes and the Ragulator complex control cell migration. Our understanding of both regulators (lysosome and the Ragulator complex) is rudimentary and this work indicates that Ragulator controls myosin phosphatase for uropod retraction. Some experimental support is still needed for a complete testing of the proposed pathway. In general, however, this is a thorough and careful study and enthusiasm is high.

Major comments

1) Western blots throughout the manuscript should be presented with a graph showing the average intensity values for at least 3 independent experiments.

2) The lysosome localization analysis presented in Fig S1 needs to be better described. How was the data acquired? Is this a line scan? How were the values averaged in space (front-to-back) along the thickness of the cell? The data should be represented as average from x number of lines scans from x number of cells.

3) The conclusion that Ragulator controls cell migration by modulating myosin activity is somewhat preliminary (lines 171-172), as myosin activity itself is not formally tested. The interaction of Lamtor1 and MRPIP is tested. Loss of this interaction (in a Lamtor1 knockouts or complementation

with mutants that don't bind Ragulator) increases MRPIP interaction with MYPT1, which decreases p-MLC levels (Fig. 2F). But in this case, the role of p-MLC levels in the migration phenotypes are not tested. However, knocking down MRPIP is tested and shown to be required for the low dose myosin inhibitors or MLC-AA mutants should be used to phenocopy the Lamtor1 knockout phenotypes in 2e and p-MLC DD mutants can be tested to complement loss

4) The conclusion that the Ragulator complex, delivered to the uropod by lysosomes, inactivates MLCP by binding to MRPIP and permits myosin II activity (indicated by phospho-MLC (lines 214-217) needs further testing. While the Ragulator interaction is needed, a specific role for uropod-localized Ragulator in the regulation is not shown. Spatial activation/inactivation of Ragulator in the uropod with localization mutants or optogenetics is needed to test this conclusion.

Minor comments dealing with ease of reading and interpretation:

1) The description of published data on the function of Lamtor1 in macrophages is somewhat confusing (lines 62-65). It is unclear how the effects of lipid metabolism and cytokine production affect cell polarization, in at all.

2) The definition of "peripheral lysosomes" (line 66-67 and) is confusing with respect to common terms used in the migration field. Usually, "peripheral" is used to mean at the cell edge. Here, it appears the authors are using the term to refer to anything outside of the peri-nuclear region, as described in the Figure 1 figure legend.. This distinction needs to be made more clear.

3) The description of published data in fibroblasts (lines 70-71) is too much of an over-simplification. As written, it currently implies that Lamtor loss induces abnormal lysosome localization, which and this abnormal lysosome localization reduces cell migration, specifically due to loss of integrin recycling and reduced adhesion dynamics. While Lamtor ½ are retained lysosomes, the references don't show that this is cause of the observed reduction in motility.

4) The description of the collagen matrix (lines 77-78) also oversimplifies the modeled 3D environment. Which 3D environment do the authors mean? The ECM composition, stiffness, and structural organization varies greater from tissue to tissue.

5) The introduction of MRPIP and MYPT1 is not sufficient (lines 164). An explanation of how MRPIP inhibits MYPT1 and how MYPT1 inhibits p-MLC is needed.

6) A better explanation of the assay for DC recruitment after using FITC-microbead-bearing DCs (line 181). The explanation should include what the assay tests and how the data described are interpreted to reach the authors conclusions.

7) The description of the expression levels of co-stimulatory molecules, in wildtype and Lamtor1-/- BMDCs is confusing (Supplementary Figure 9d, line 189). Expression appears higher in the figure rather than lower. What does this mean?

Reviewer #3 (Remarks to the Author):

The authors investigated the role of the Ragulator complex in respect to dendritic cell migration in conditional knockout mice. The authors found that Ragulator is required for dendritic cell migration. Mechanistically, the Ragulator complex component lamtor1 interacts with myosin phosphatase Rho-interacting protein (MPRIP), and disrupts the interaction between MPRIP and the myosin light chain phosphatase (MLCP) phosphatase resulting in the dephosphorylation of myosin light chain MLC. Some concerns exist, refer to comments below.

Major:

The authors ruled out the involvement of mTORC1. Ragulator is also known to regulate MAPK pathway, and MAPK pathway has also been implicated in MLC phosphorylation. Does the MAPK pathway play a role here?

Co-localization was shown between lamtor1 and MPRIP. What about MYPT1? The authors claim that Lamtor1 disrupts the interaction between MPRIP and MYPT1. Immunofluorescence of both MPRIP and MYPT1, in the presence and absence of Lamtor1 is needed.

Figure 1 B, C shows a similar cellular distribution pattern of lysosome and Lamtor1. It would be helpful if LAMP1 and Lamtor1 can be co-stained during non-polarized state and a polarized state to make sure colocalization of LAMP1 and Lamtor1 is consistent during the two states.

Figure 1D, in the Lamtor1 knockout background, there is a reduced response to chemotaxis (CCL19 and CCL21) compared to wildtype dendritic cells. Do Lamtor1 ^{-/-} DC cells still have the same number of lysosomes as wildtype DC cells? Maybe it would be helpful quantify the number of lysosomes in those two types of cells.

Figure 2D and E. Other lysosomal proteins need to be probed for like LAMP1, etc. It is possible that lysosomes are just being pulled out.

alpha-4E mutant doesn't bind to MPRIP, or rescue p-MLC or DC migration (Figures 3A,3D,3E). But in Figure 3C, alpha-4E mutant appears to still reduce the binding between MYPT1 and MPRIP? Please explain.

Figure 3F - Please show extent of MPRIP knock-down in Lamtor1-KO-THP1. What about MYPT1 expression level in these cells?

Figures 3A, B - it's interesting that the author can abolish Lamtor1 interaction with MPRIP. But is MPRIP still colocalizing with lysosome? Maybe is helpful to blot for MPRIP interaction with other lysosomal marker, or try to co-stain MPRIP with lysosome marker to examine the colocalization by using confocal imaging.

Figure 4B - can increased abundance of DCs itself negatively impact on mobility of DCs?

Do the authors have evidence that Lamtor1 (rather than other components of Ragulator) directly interacts with MPRIP?

Can phenotypes in Figure 4 be rescued by Lamtor1?

Figure 4 - The authors used the subset of migratory DCs for confirming DCs trafficking and immunity adaptation. Please describe how the subset of migratory DCs were obtained from total DCs.

Does MPRIP interact with other Ragulator subunits including Lamtor2/4 and Lamtor3/5 or Rag GTPase?

Please describe why Lamtor1 mutants ($\Delta T1$, Met1-Ser144; $\Delta T2$, Met1-Gln114; $\Delta T3$, Met1-His41; and $\alpha 4E$) were generated. Are those sites known to be interacted with Lamtor1 and other Ragulator components?

What is % input in chemotaxis experiments evaluated by Transwell assay?

Minor:

Western blot data is needed to verify over expression, knockdown or knockout. For example, Figure 2F: blot for Lamtor1, Figure 3F: blot for MPRIP, etc.

line 87: "Hence, we focused on the Ragulator complex because of the similar distribution ...". The rationale of focusing on Ragulator is very farfetched, because any lysosomal protein will have similar distribution here.

Figures 1B, 1C - What is a stimulator for polarized cells?

Figure 1D legend - "the indicated concentrations of CCL19 (left) or CCL21 (right) in a Transwell assay system" should be changed to match the arrangement of figure.

Figure 1J - Authors showed the expression of p-MLC in Lamtor1 WT and KO of DC cells. What expression level of p-MLC when those cells (Lamtor1 WT and KO) exposed in CCL19 or CCL21?

Figure 1J, show the blots of Lamtor1 knockout and LAMP1 level.

Figure 2D use endogenous IP to demonstrate Lamtor1 and MPRIP interaction in a physiological relevance.

Figure 2G, show blots of Lamtor1.

Figure 3d, show Lamtor1-FLAG and $\alpha 4E$ -FLAG expression.

Figure 3f, show MPRIP expression level

Draw out a working model.

Point-by-point replies to Reviewers

First, we would like to thank the reviewers for their many constructive comments and suggestions. Please note that we have modified the revised manuscript according to the preferred article format for *Nature Communications*. So, while the format appears different, we were careful to retain the content and logic of the original submission.

#1 Response to Reviewer 1

Nakatani et al. present a comprehensive, engaging research paper that is of interest to the immunology community, bridges a knowledge gap, and highlights potential clinical applications.

Broadly, the group investigated the role of the lysosome-bound Ragulator complex in the motility of dendritic cells. They show that Lamtor-1, one of the components of the pentameric Ragulator, colocalizes with lysosomes in the uropod of polarised bone marrow derived dendritic cells (BMDCs). Lamtor-1 also colocalized with phosphorylated myosin light-chain (p-MLC). Both MLC phosphorylation and BMDC migration were impaired in Lamtor-1^{-/-} BMDCs.

The group further showed that Lamtor-1, when it is able to form the full Ragulator complex, coprecipitates and interacts with MPRIP. MPRIP was previously shown to anchor the MYPT1 subunit of MLC phosphatase to the actin-myosin filament, effectively reducing MLC phosphorylation and thus contractility. Nakatani et al. propose that the Ragulator complex binds to MPRIP and interferes with the MPRIP-MYPT1 interaction, thus blocking the recruitment of MLC phosphatase and resulting in more p-MLC and increased contractility. They also confirm that Lamtor-1 is required for DC migration in vivo.

Interestingly, despite Ragulator acting as a scaffold for mTORC1, the addition of mTORC inhibitors had no effect on DC motility or the interaction between Lamtor-1 and MPRIP, and S6K phosphorylation was not affected in the Lamtor-1 knockout.

The paper is well-written and presents a logical story. However, there are several points that require clarification or elaboration.

Thank you for your valuable comments and for appreciating that our findings provide a conceptual bridge between cell morbidity and actomyosin contraction machinery. We have responded to your specific comments below.

1. Firstly, to strengthen the title and the conclusions of this paper, I suggest that the main experiment be

repeated with macrophages or other immune cells. The authors refer to Lamtor1's relevance in macrophages in the introduction (line 62), but then focus on DCs without a clear justification – monocytes, macrophages, T-cells and neutrophils have all been shown to move at the same velocity as DCs. DCs use different migration strategies than other immune cells (Nature Reviews Molecular Cell Biology volume 20, pages738–752 (2019). Repeating, for instance, the immunofluorescence study with macrophages could either broaden the scope of the findings or highlight a new DC-specific mechanism depending on the results. This would also justify the authors' claim in line 235 that Ragulator is “indispensable for leukocyte trafficking”. The conclusion also states that the newly discovered Ragulator/MPRIIP interaction may provide a target for combating cancer metastasis, which is not mentioned or explained anywhere else in the text – the authors should justify this claim.

1. The reviewer expressed concern that Ragulator complex–MPRIIP-mediated actomyosin contraction might also be important for immune cells other than DCs, as other leukocytes utilize the same mode of morbidity. To address this concern, we examined *in vitro* and *in vivo* neutrophil trafficking and compared migration efficiencies between WT and LysM-Cre-Lamtor1^{flox/flox} mice, in which Lamtor1 expression was almost absent in neutrophils (**Response Figure 1–1a**). Transwell assays revealed that the chemokine response to fMLP was weaker in LysM-Cre Lamtor1^{flox/flox} neutrophils isolated from bone marrow than in WT neutrophils (**Response Figure 1–1b**). Also, when LPS was administered to the peritoneum of WT or LysM-Cre Lamtor1^{flox/flox} mice to evaluate *in vivo* neutrophil mobilization, the infiltrating neutrophils in the peritoneal fluid 4 hours later were less abundant in LysM-Cre-Lamtor1^{flox/flox} mice than in WT mice. (**Response Figure 1–1c**). These results suggested that Lamtor1 also plays a crucial role in neutrophil trafficking. On the other hand, the significance of Lamtor1 in lymphocyte trafficking remains unclear, as we have already terminated the colony of CD4-Cre Lamtor1^{flox/flox} mice and failed to establish Lamtor1-deficient Jurkat cells using CRISPR technology. However, we believe that it was important to clarify that Lamtor1 is involved in the trafficking of DCs as well as neutrophils.

Response Figure 1–1

To emphasize the importance of Lamtor1 in immune cell trafficking, we included this figure as **Figure 9c–e**, and the following text was added (**page 18**).

Impaired trafficking in *Lamtor1*^{-/-} neutrophils

Finally, to investigate the motility of leukocytes other than DCs, we generated LysM-Cre × Lamtor1^{fllox/fllox} mice, in which Lamtor1 was specifically knocked out in neutrophils and macrophages (**Figure 9c**). The fMLP-induced migration of neutrophils was lower in Lamtor1^{-/-} neutrophils isolated from bone marrow than in WT neutrophils (**Figure 9d**). In addition, we injected LPS into the peritoneum of WT and LysM-Lamtor1^{fllox/fllox} mice and counted the number of infiltrated neutrophils in the peritoneal fluid 6 hours after injection by flow cytometry. The percentage of total peritoneal cells that were CD11b⁺Ly6G⁺ and the absolute number of neutrophils were considerably reduced (**Figure 9e**). These results indicate that Lamtor1 is indispensable for cell motility not only in DCs, but also in neutrophils.

On the other hand, we have removed the discussion of the role of Lamtor1 in cancer cell migration because this paper does not show any relevant data.

2. Although the authors did not find a link between their results and mTORC1 activity, they themselves state that the findings on mTORC1's role in leukocyte activity have thus far been controversial. The authors may wish to consider including information on the amino acid content of their chosen medium (RPMI), and could repeat the mTORC1 inhibition experiments under more nutrient-rich conditions (e.g. in DMEM) or minimal conditions.

2. Regarding the role of mTORC1 in DC migration, the reviewers asked whether amino acid content affects mTORC1 inhibition and recommended that we compare DC migration between different media, e.g., DMEM (higher amino acid content) and RPMI (lower amino acid content). To address this concern, we conducted a DC migration assay in DMEM medium (**Response Figure 1–2a**) and amino acid-free RPMI (**Response Figure 1–2b**) in order to investigate the effect of rapamycin on DC migration in the presence of different amino acid contents. In both cases, DC migration did not change despite apparent inhibition of S6K phosphorylation (**Response Figure 1–2a, b**), suggesting that the contribution of mTORC1 activity to DC migration is not related to amino acid nutritional status.

Response Figure 1–2

To support the conclusion that mTORC1 is not involved in DC migration in Figure 3, we have included this figure as supplementary **Figure 3a, b** and added the following text. (**page 10**).

When DCs were treated with rapamycin and torin1 (an mTORC1/mTORC2 inhibitor) at doses that completely inhibited phosphorylation of S6K, DC motility in response to CCL19 was not affected (**Figure 3a, b**), regardless of amino acid concentration (**Supplementary Figure 3a, b**).

3. The authors present a large amount of western blot data. However, the methods and figure legends do not sufficiently describe the detection methods that were employed. These should be added for reproducibility. Additionally, none of the blots were quantified. As western blotting is often used to show relative amounts, quantification is often not required. However, to strengthen certain claims quantification would be useful; for example for Figures 2G and 3C.

3. The reviewer requested quantification of Western blotting data. We fully agree with this request. Accordingly, for Figures 1j, 2c, 2f, 2g, 3c, and 3d, we conducted at least three independent western blotting examinations, quantified the protein concentration in SDS-PAGE gel bands using the ImageJ software, and performed statistical analysis by Student's t-test. Bar graphs show means ± standard deviation (**Response Figure 1–3**).

These quantitative graphs were posted next to the images of western blots (Figures 1J, 2c, 2f, 2g, 3c, and 3d in the previous manuscript were moved to Figures 2g, 3d, 5a, 6a, 7d, and 7e, respectively). Also, we will describe the quantitative method in the Materials and Methods section (**page 40**).

Response Figure 1–3

4. There is also a lack of clarity regarding the polarized state of DCs. The authors generally use CCL19, but include CCL21 in Figure 1D. DC motility may vary in response to different chemokines (see e.g. Schumann et al. (2010) in Immunity), so it may be interesting to repeat some experiments with CCL21 as the polarizing agent. Regardless, if “polarized” always specifically refers to “CCL19-treated”, this should be clarified. It is unclear whether the cells in Figure 1J were treated in some way to induce polarization.

4. The reviewer argued that the method for polarization of cells needs to be carefully explained. In *in vitro* experiments using Transwell assays, DC migration was induced by both CCL19 and CCL21, but the DC locomotion assay in collagen gel was performed using CCL19 exclusively.

Therefore, I would like to write the polarization agent as “CCL19-treated DCs” in these experiments.

Also, the reviewer wanted to know which polarization agent was used in the Figure 1J experiment. Before the experiment shown in Figure 1J, we investigated whether CCL19 can induce phosphorylation of MLC. However, increased MLC phosphorylation was not observed in WT or Lamtor1^{-/-} DCs 10 min after CCL19 administration, and MLC phosphorylation was considerably reduced in Lamtor1^{-/-} DCs irrespective of CCL19 administration (**Response Figure 1–4**). Therefore, we showed the result without a polarization agent and quantitatively evaluated MLC phosphorylation under the non-stimulated condition.

Response Figure 1 – 4

We added the following to the text (**page 9**).

Additionally, MLC phosphorylation was lower in *Lamtor1*^{-/-} DCs than in WT DCs even under non-stimulated conditions (**Figure 2g**).

5. On another small note, there are some layout issues. For example, the figure legend for Figure 1D refers to the “left” and “right” graphs, when the two graphs are in fact stacked vertically. Additionally, the placement of Supplementary Figure 1B is odd in terms of the paper’s chronology, as it is first referred to very late in the text, only after Supplementary Figure 4. Supplementary Figure 1A shows the merging of the two stains shown in Figures 1B and 1C, and since the authors refer to colocalization, they may want to consider including the merged image in the main text, as this is more convincing. Finally, I would consider rephrasing that DCs are “professional cells” (line 175); perhaps changing it to the implied “professional antigen-presenting cells”.

5. The reviewer pointed out that the figures were in an inappropriate order and recommended a way to present the images. We thank the reviewer for pointing out our mistakes. We corrected Figure 1d and the legend. Also, as the reviewer recommended, we posted the merged figure showing colocalization of Lamtor1 and LAMP1 in **Figure 1d**. Also, we changed the text from "DC are professional cells" to "DCs are professional antigen-presenting cells."

6. There is a small lack of information regarding the FACS assay of dendritic cells, which are classified as “migratory” based only on their expression of MHC-II. There are multiple diverse subsets of DCs with a variety of different markers. For more accuracy, CD103 should be included. The authors may want to consider including or explaining their gating strategy as well.

6. The reviewer had a concern about the definition and gating strategy for migratory DCs. In the original manuscript, a very large population of CD11c⁺MHC II^{hi} cells were defined as migratory DCs. We reassessed the number of CD11c⁺CD103⁺ DC subsets in the cutaneous draining lymph nodes and dermis of WT and CD11c–Lamtor1^{flox/flox} mice (**Response Figure 1–5a, b**). Consistent with the CD11c⁺MHC II^{hi} population, CD11c–Lamtor1^{flox/flox} mice had a lower number of CD11c⁺CD103⁺ DCs in the cutaneous draining LNs relative to WT mice, and tended to have a higher number in the dermis. Both gating methods confirmed that Lamtor1-deficient DCs impaired trafficking to the lymph nodes.

Response Figure 1 – 5

We added the results regarding CD103⁺DCs to Figure 8a, b. In addition, we added the following text on **page 16** and described the FACS gating strategy in the legend of Figure 8a (**page 30**).

CD11c–*Lamtor1*^{flx/flx} mice had a reduced number of CD11c⁺MHCII^{hi} DCs and CD11c⁺CD103⁺DCs, both of which are representative of migratory DCs, in skin-draining LNs (dLNs)^{27, 41} (**Figure 8a**), although they had abundant DCs in the skin itself (**Figure 8b**).

Figure 8: Impaired *in vivo* DC trafficking in DC-specific *Lamtor1*^{-/-} mice

a) Migratory DC subset in inguinal LNs. Single-cell suspension from inguinal LNs was stained for CD11c and MHC II, and migratory DCs (red rectangle) were counted by flow cytometry (left). Population (upper) and absolute number (lower) of CD11c⁺MHC II^{hi} DCs (middle), and CD11c⁺CD103⁺ DCs (right) of WT (white bar) and CD11c–*Lamtor1*^{flx/flx} (black bar) mice are shown.

Response to Reviewer 2

In the manuscript “The lysosomal Ragulator complex plays an essential role in leukocyte trafficking by activating myosin II” by Nakatani et al, the authors describe new mechanistic understanding of how lysosomes and the Ragulator complex control cell migration. Our understanding of both regulators (lysosome and the Ragulator complex) is rudimentary and this work indicates that Ragulator controls myosin phosphatase for uropod retraction. Some experimental support is still needed for a complete testing of the proposed pathway. In general, however, this is a thorough and careful study and enthusiasm is high.

Thank you for your valuable suggestions and favorable comments about our work. We have responded to your specific comments below.

Major comments:

1) Western blots throughout the manuscript should be presented with a graph showing the average intensity values for at least 3 independent experiments.

1. The reviewer requested quantification of Western blotting data. We fully agree with this request. As mentioned above (**Response to Reviewer 1–3**), we conducted at least three independent western blotting examinations, quantified the concentration of protein in SDS-PAGE gel bands using the ImageJ software, performed statistical analysis using Student’s t-test. Bar graphs show mean \pm standard deviation.

These quantitative graphs were posted next to the images of Western blots (Figures 1J, 2c, 2f, 2g, 3c, and 3d in the previous manuscript were moved to Figures 2g, 3d, 5a, 6a, 7d, and 7e, respectively). Also, we will describe the quantitative method in the Materials and Methods section (**page 40**).

2) The lysosome localization analysis presented in Fig S1 needs to be better described. How was the data acquired? Is this a line scan? How were the values averaged in space (front-to-back) along the thickness of the cell? The data should be represented as average from x number of lines scans from x number of cells.

2. Reviewer was concerned about how the fluorescence images were taken and co-localization was evaluated. All fluorescence images were obtained by confocal microscopy. To evaluate co-localization, a Z-stack image was acquired by optimizing the pinhole size according to the objective lens ($\times 63$), and representative slice images are shown in the manuscript. The imaging procedure is described in Materials and Methods (**page 41**). To make it easier to understand, we added a merged diagram

showing co-localization of Lamtor1 and LAMP1 to **Figure 1d (Response Figure 2-1)**.

Response Figure 2 – 1

3) The conclusion that Ragulator controls cell migration by modulating myosin activity is somewhat preliminary (lines 171-172), as myosin activity itself is not formally tested. The interaction of Lamtor1 and MRPIP is tested. Loss of this interaction (in a Lamtor1 knockouts or complementation with mutants that don't bind Ragulator) increases MRPIP interaction with MYPT1, which is decreases p-MLC levels (Fig. 2F). But in this case, the role of p-MLC levels in the migration phenotypes are not tested. However, knocking down MRPIP is tested and shown to be required for the Low dose myosin inhibitors or MLC-AA mutants should be used to phenocopy the Lamtor1 knockout phenotypes in 2e and p-MLC DD mutants can be tested to complement loss.

3. The reviewer implied that assessing p-MLC phosphorylation alone could not conclude, and requested to investigate whether the impaired motility of Lamtor1-deficient cells could be rescued by activation or inhibition of actomyosin. To answer this, we investigated whether the motility of Lamtor1-KO-Lamtor1-Full THP1 cells, whose motility was comparable to that of parental WT THP1, would be reduced by treatment with Blebbistatin, an actomyosin inhibitor. We determined that the motility of Lamtor1-KO-Lamtor1-Full THP1 cells was decreased to that of Lamtor1-KO-THP1 (**Response Figure 2–2a**). In addition, Lamtor1-KO THP1 transduced with the active form of MLC (DD-MLC; replaced Thr18Asp and Ser19Asp) (**Response Figure 2–2b**) had greater motility than Lamtor1-KO THP1 (**Response Figure 2–2c**) although the recovery was only partial relative to the parental WT-THP1. These results indicate that the Ragulator complex regulates cell motility by regulating MLC II-mediated actomyosin contraction.

Response Figure 2 – 2

The results were included in **Figure 6a–c**, and the following sentences were added to the text (**page 13**).

Lamtor1-mediated motility is driven by myosin II activity

Next, we investigated whether myosin II–dependent actomyosin is important for Lamtor1-mediated motility. MLC phosphorylation, which was reduced in Lamtor1-KO-THP1, was restored to the same level as in WT-THP1 by add-back of Lamtor1 (**Figure 6a**). In addition, chemotaxis in response to CCL2 was reduced in Lamtor1-KO-THP1, yet restored to the same level as in WT-THP1 in Lamtor1-KO-Full-THP1, and the restored motility in Lamtor1-KO-Full-THP1 was diminished by treatment with blebbistatin, a myosin II inhibitor (**Figure 6b**). Furthermore, motility, which was reduced in Lamtor1-KO THP1, was restored by forced expression of a constitutively active form of MLC (MLC-DD, carrying the Thr18Asp and

Ser19Asp mutations)⁴⁰ in Lamtor1-KO-THP1 (**Figure 6c**).

4) The conclusion that the Ragulator complex, delivered to the uropod by lysosomes, inactivates MLCP by binding to MPRIP and permits myosin II activity (indicated by phospho-MLC (lines 214-217) needs further testing. While the Ragulator interaction is needed, a specific role for uropod-localized Ragulator in the regulation is not shown. Spatial activation/inactivation of Ragulator in the uropod with localization mutants or optogenetics is needed to test this conclusion.

4. The reviewer had a concern about the specific role of the Ragulator complex in the uropod, and suggested using photoactivation or optogenetics to address this issue. Unfortunately, due to technical and equipment limitations, we were not able to perform photoactivation and optogenetics to achieve spatiotemporally regulated protein expression or depletion. Therefore, we tried to answer the question by increasing lysosome transport to the plasma membrane. To this end, we introduced the active form of Arl8b (Arl8b-CA) into WT THP1, as the Arl8b-BORC system is involved in the trafficking of lysosomes to the periphery (*Dev Cell.* 2015,33(2):176-88, *J Cell Biol.* 2017,216(12):4199-4215), and examined the motility of Arl8b-CA-THP1. Lysosomes were preferentially localized close to the plasma membrane in Arl8b-CA-THP1 cells (**Response Figure 2–2a**). However, these cells did not respond to chemokines (**Response Figure 2–2b**), presumably because cell polarity might be disturbed by forced expression of Arl8b-CA. Therefore, we abandoned this strategy.

Response Figure 2 – 3

Instead, to further clarify the relationship between cell motility and the interaction between the Ragulator complex and MPRIP at the trailing, we are trying to identify the region of MPRIP that binds to the Ragulator complex (**Response Figure 2–3a**). To date, we have identified the N-terminal region of MPRIP as critical for binding with the Ragulator complex (**Response Figure 2–3b, c**); moreover, both Lamtor1 and Lamtor2 seem to be important (**Response Figure 2-3d**). These data are quite preliminary and we would like to confirm the findings before reporting them in a publication. Using MPRIP mutant cell lines and mice and knock-out cell lines of each Ragulator complex

component, we will elucidate the detailed interaction between the Ragulator complex and MPRIP. We plan to address this issue thoroughly in a subsequent study.

Response Figure 2 – 4

Minor comments:

1) The description of published data on the function of Lamtor1 in macrophages is somewhat confusing (lines 62-65). It is unclear how the effects of lipid metabolism and cytokine production affect cell polarization, in at all.

1. The reviewer argued that the function of Lamtor1 in macrophages was not well explained. It was not possible to give a longer explanation due to word count limits. In the Introduction, we explained in greater detail the role of Lamtor1 in regulating lipid metabolism and cytokine production (**page 4**).

The lack of Lamtor1 leads to reduced phosphorylation of S6K and lower levels of the LXR ligand, 25-hydroxycholesterol, resulting in expression of LXR-dependent genes that are important for differentiation of M2 macrophages. Also, pro-inflammatory cytokine production is elevated in

Lamtor1^{-/-} macrophages due to the increase in nuclear translocation of transcription factor EB (TFEB)⁹.

2) The definition of “peripheral lysosomes” (line 66-67 and) is confusing with respect to common terms used in the migration field. Usually, peripheral” is used to mean at the cell edge. Here, it appears the authors are using the term to refer to anything outside of the peri-nuclear region, as described in the Figure 1 figure legend. This distinction needs to be made more clear.

2. The reviewer argued that the term "peripheral lysosomes" was not appropriate because this term is normally used to refer to lysosomes located at the edges. As the reviewer pointed out, lysosomes close to the plasma membrane should be called "peripheral lysosomes." In primary DCs, the morphology of cells was so complex and rapidly altered that it was very difficult to analyze lysosome localization statistically. Also, because lysosomes distributed in the peripheral region were located slightly inside the plasma membrane, we adopted a method in which we divided the inside and outside based on a boundary drawn 5 µm outside the nucleus. We used the term "peripheral lysosomes" because we thought it would be easier for readers to visualize, but as the reviewer pointed out, terminology should be strict. Accordingly, we replaced "peripheral lysosome" with "lysosomes outside the perinuclear region," “lysosomes in the peripheral region,” or “peripherally transported lysosomes.”

3) The description of published data in fibroblasts (lines 70-71) is too much of an over-simplification. As written, it currently implies that Lamtor loss induces abnormal lysosome localization, which and this abnormal lysosome localization reduces cell migration, specifically due to loss of integrin recycling and reduced adhesion dynamics. While Lamtor ½ are retained lysosomes, the references don't show that this is cause of the observed reduction in motility.

3. The reviewer argued that the explanation of bibliographies that showed the Lamtor1's roles in MEF was oversimplified. Due to the word count limit, we could not explain the role of the Ragulator complex in mesenchymal cell movement. In the revised text, we added an explanation of the involvement of the lysosome and the Ragulator complex in cell movement in the text as follows (**page 5**).

Mesenchymal cells, such as fibroblasts, move via lamellipodia and filopodia through actin polymerization, binding with ECM by integrin, subsequent pulling at the leading edge, and detachment from the ECM mediated by integrin recycling. Lysosomes are involved in this process BORC, a multi-subunit complex, recruits the GTPase Arl8 to the lysosomal membrane. Arl8, in turn, promotes the movement of lysosomes to the cell periphery, resulting in cell spreading and

motility. Additionally, Lamtor1-deficient MEFs exhibit impaired cell motility because of impaired cycling of $\beta 1$ integrin due to an abnormal distribution of lysosomes resulting from impaired interaction with BORC. Also, Lamtor2/3 moves to the cell periphery via Rab7-positive late endosomes and promotes maturation of focal adhesions via association with IQGAP1. Therefore, lysosomes and Ragulator complexes play roles in mesenchymal cell migration by regulating integrin-dependent motility mechanisms.

4) The description of the collagen matrix (lines 77-78) also oversimplifies the modeled 3D environment. Which 3D environment do the authors mean? The ECM composition, stiffness, and structural organization varies greater from tissue to tissue.

4. The reviewers argued that the description of the collagen matrix in the 3D environmental model was oversimplified. As the reviewer pointed out, the density and stiffness of ECM vary from tissue to tissue (*Nat Cell Biol.* 2018, 20 (1):8-20). However, previous reports have optimized the concentration of type I collagen for BMDC locomotion by assessing the morphological changes and motility of BMDCs. To mimic the 3D environment for DC locomotion experiments, these studies utilized 1.5–3 mg/ml type I collagen (*Nature.* 2008, 453(7191):51-5). Accordingly, we also used 2 mg/ml type I collagen. We altered the text as follows (**page 9**).

... *Lamtor1*^{-/-} DCs that migrate in response to CCL19 in a 3D collagen matrix consisting of 2 mg/ml Type I collagen to mimic the physiological 3D environment of DCs,

5) The introduction of MRPIP and MYPT1 is not sufficient (lines 164). An explanation of how MRPIP inhibits MYPT1 and how MYPT1 inhibits p-MLC is needed.

5. The reviewer requested a clearer explanation of how MRPIP inhibits MYPT1 and how MYPT1 inhibits p-MLC. Due to the limited number of characters, we provided a less detailed explanation of the mechanism by which MRPIP dephosphorylates MLC via MYPT1. MRPIP (also known as p116^{Rip}) was originally identified as a RhoA-interacting protein that associates with actin. The myosin phosphatase complex (MLCP), which consists of catalytic subunit PP1c δ , M20, and MYPT1 (also known as MBL), is anchored to actin by an interaction between the coiled-coil region of MRPIP and the Z region of MYPT1, leading to dephosphorylation of MLC. We explain this in the Introduction as follows (**page 6**).

Phosphorylation of the myosin light chain (MLC) is regulated by the balance between Ca²⁺/calmodulin-mediated MLC kinase (MLCK) and MLC phosphatase (MLCP). MLCP is a

heterotrimer consisting of myosin phosphatase–targeting subunit 1 (MYPT1), a 20-kDa small subunit (M20), and a catalytic subunit of the type I protein serine/threonine phosphatase family (PP1c δ). MLCP is anchored on actin–myosin bundles by myosin phosphatase–Rho interacting protein (MPRIIP, also known as p116^{Rip}), and MLCP activity is regulated by RhoA via suppression of MYPT1 phosphorylation, which inactivates the catalytic activity of PP1c δ .

6) A better explanation of the assay for DC recruitment after using FITC-microbead-bearing DCs (line 181). The explanation should include what the assay tests and how the data described are interpreted to reach the authors conclusions.

6. The reviewer requested a clearer explanation of the FITC microbead assay for DC mobilization. In general, the FITC skin sensitization assay using the FITC isomer can assess the homing of DCs to skin-draining LNs, but resident DCs located in the LNs can capture FITC isomers delivered by lymphatic flow. Therefore, we performed anatomical analysis of lymphatic vessels as well as an *in vivo* DC migration assay using FITC microbeads, which cannot be delivered by lymphatic flow. CD11c–*Lamtor1*^{−/−} mice had normal lymphatics and reduced numbers of FITC+ DCs in LNs in both the FITC-skin painting and FITC-microbeads injection assays, indicating impaired *in vivo* DC-trafficking under *Lamtor1* deficiency. We provided the following explanation in the text (**page 16**).

We then performed subcutaneous injections of Evans blue dye to evaluate the lymphatic flow system and rule out the possibility of impaired delivery of FITC-isomers. In addition, we performed the FITC microbead injection assay to evaluate the ability of *in vivo* DC migration more strictly, as microbeads cannot be delivered by the lymphatic flow. CD11c–*Lamtor1*^{flox/flox} mice exhibited no anatomical abnormality of the lymphatic system (**Supplementary Figure 7a**), and had a lower number of FITC-microbead-bearing DCs in dLNs than WT mice (**Figure 8d**).

7) The description of the expression levels of co-stimulatory molecules, in wildtype and *Lamtor1*^{−/−} BMDCs is confusing (Supplementary Figure 9d, line 189). Expression appears higher in the figure rather than lower. What does this mean?

7. The reviewer said that the consequences of the expression of co-stimulatory molecules were confusing. The expression levels of co-stimulatory molecules were slightly higher in *Lamtor1*-deficient DCs than in wild-type DCs. However, *in vitro* co-culture experiments using DCs and OT-II T cells revealed no difference in T-cell activation. Therefore, even though expression levels of costimulatory molecules were slightly higher, due to an undetermined mechanism, we concluded that impaired antigen-specific T cell priming in CD11c–*Lamtor1*^{flox/flox} mice were due not to impaired T-

DC interactions but to impaired DC trafficking to the lymph node. We provided the following explanation in the text (**page 17**).

Because DC maturation is important for T-cell priming, we evaluated the levels of co-stimulatory molecules, including MHC-II, CD80, CD86, and CD40. Expression of these proteins was not impaired in *Lamtor1*^{-/-} BMDCs with or without LPS stimulation, but was slightly higher than in WT BMDCs (**Supplementary Figure 7d**), suggesting that DC maturation was not impaired in *Lamtor1*^{-/-} DCs. To assess the *in vitro* T-cell priming ability of DCs, we performed co-cultures of OVA protein- or OT-II peptide-pulsed DCs and OT-II T-cells. OVA protein- or OT-II peptide-pulsed *Lamtor1*^{-/-} DCs promoted *in vitro* OT-II T-cell proliferation to the same extent as WT DCs, even though lysosomes are involved in endocytosis and antigen presentation (**Supplementary Figure 7c**). These results indicate that impaired DC transport in *Lamtor1*^{-/-} DCs, rather than impaired T-DC interaction or DC activation, caused impaired *in vivo* antigen-specific T-cell priming.

Response to Reviewer 3

The authors investigated the role of the Ragulator complex in respect to dendritic cell migration in conditional knockout mice. The authors found that Ragulator is required for dendritic cell migration. Mechanistically, the Ragulator complex component lamptor1 interacts with myosin phosphatase Rho-interacting protein (MPRIIP), and disrupts the interaction between MPRIIP and the myosin light chain phosphatase (MLCP) phosphatase resulting in the dephosphorylation of myosin light chain MLC. Some concerns exist, refer to comments below.

Thank you for your valuable comments and suggestions. We have responded to your comments below.

Major comments:

1) The authors ruled out the involvement of mTORC1. Ragulator is also known to regulate MAPK pathway, and MAPK pathway has also been implicated in MLC phosphorylation. Does the MAPK pathway play a role here?

1. The reviewer had a concern about the involvement of the MAPK pathway in Ragulator complex-mediated DC-trafficking. As the reviewer pointed out, the Ragulator complex has been reported to act as an adapter for the MAPK pathway. However, when WT BMDCs were treated with U0126, an ERK inhibitor that completely abolished ERK phosphorylation, their response to CCL19 was not reduced (**Response Figure 3-1a**). Additionally, MLC phosphorylation made no difference, irrespective of U0126 treatment (**Response Figure 3-1b**), suggesting that activation of MEK is not essential for DC migration.

Response Figure 3-1

We included these results in **Figure 3e, f** and added the following description to the text (**page 11**).

In addition, the Ragulator complex is involved in the MEK pathway because the Lamtor2/3

complex is a scaffold for MEK. However, DCs treated with the MEK inhibitor U0126 did not decrease CCL19-induced cell motility even at high drug concentrations (**Figure 3e**). Moreover, phosphorylation of MLC did not differ between U0126-treated DCs and untreated DCs (**Figure 3f**). These results suggested that the MEK-dependent pathway is dispensable for DC migration.

2) Co-localization was shown between lamtor1 and MPRIP. What about MYPT1? The authors claim that Lamtor1 disrupts the interaction between MPRIP and MYPT1. Immunofluorescence of both MPRIP and MYPT1, in the presence and absence of Lamtor1 is needed.

2. The reviewer was interested in knowing whether co-localization of MPRIP and MYPT1 differed in the presence and absence of Lamtor1. We compared the localization of MYPT1 in WT and Lamtor1-KO THP1 cells. In Lamtor1-KO THP1, MYPT1 was detected in the uropod (**Response Figure 3–2a**), where it co-localized with MPRIP (**Response Figure 3–2b**). By contrast, WT THP1 contained less MYPT1 in the uropod (**Response Figure 3–2a**), and co-localization with MPRIP was also reduced (**Response Figure 3–2b**). This result supports the finding that the Ragulator complex inhibits the interaction between MPRIP and MYPT1 in the uropod.

Response Figure 3–2

We added these results to **Figure 5b** and briefly described this observation in the text (**Page 12**).

In addition, we compared localization of MPRIP and MYPT1 in WT and Lamtor1-KO THP1 cells by confocal microscopy. Interestingly, in WT THP1, localization of MYPT1 to the uropod was reduced, whereas in Lamtor1-KO THP1, MYPT1 was present in the uropod, and it co-localized with MPRIP in the uropod to a greater extent than in WT THP1 (**Figure 5b**).

3) Figure 1 B, C shows a similar cellular distribution pattern of lysosome and Lamtor1. It would be helpful if LAMP1 and Lamtor1 can be co-stained during non-polarized state and a polarized state to make sure colocalization of LAMP1 and Lamtor1 is consistent during the two states.

3. The reviewer requested that we display the co-localization of Lamp1 and Lamtor1 both at steady state and in the polarized state. Regardless of cell state, Lamp1 and Lamtor1 were co-localized (**Response Figure 3-3**).

Response Figure 3–3

We added this image to **Figure 1d** and the following sentence to the text (**page 8**).

... and Lamtor1 and Lamp1 were co-localized regardless of cell polarization (**Figure 1d**).

4) Figure 1D, in the Lamtor1 knockout background, there is a reduced response to chemotaxis (CCL19 and CCL21) compared to wildtype dendritic cells. Do Lamtor1 ^{-/-} DC cells still have the same number of lysosomes as wildtype DC cells? Maybe it would be helpful quantify the number of lysosomes in those two types of cells.

4. The reviewer was interested to know whether the number and distribution of lysosomes differed between WT and Lamtor1^{-/-} DCs. The expression levels of Lamp1 in WT and Lamtor1^{-/-} BMDCs were evaluated by western blotting; there was no noticeable difference between the genotypes (**Response Figure 3-4a**). Regarding the distribution of lysosomes, lysosomes were preferentially localized outside the perinuclear region in Lamtor1^{-/-} BMDCs regardless of whether the cells were polarized state or non-polarized (**Response Figure 3-4b**), as described previously for fibroblasts (*EMBO J.* 2009; 28(5):477-89). These results indicate that the impaired cell motility in Lamtor1^{-/-}

cells is not due to a reduction in the number of lysosomes or the absence of lysosomes in the uropod.

We added these results to **Figure 2h, 2i** and provided the following explanation in the text (**page 10**).

Furthermore, lysosomes tended to be distributed to the peripheral region in non-polarized *Lamtor1*^{-/-} DCs, as observed previously in fibroblasts, and also localized at the uropod of polarized *Lamtor1*^{-/-} DCs (**Figure 2h**). The expression level of LAMP1 protein in *Lamtor1*^{-/-} DCs was comparable to that in WT DCs (**Figure 2i**). These results suggest that impaired DC migration was not due to impairment of lysosome generation or distribution to the uropod.

5) Figure 2D and E. Other lysosomal proteins need to be probed for like LAMP1, etc. It is possible that lysosomes are just being pulled out.

5. The reviewer was concerned that lysosomal proteins such as LAMP1 might be contaminating the immunoprecipitation of Lamtor1. To rule out this possibility, we investigated whether LAMP1 was

Response Figure 3 – 5

present in the immunoprecipitated samples pulled down by Lamtor1-FLAG antibody. Lamp1 was not detected (**Response Figure 3–5**).

6) alpha-4E mutant doesn't bind to MPRIP, or rescue p-MLC or DC migration (Figures 3A,3D,3E). But in Figure 3C, alpha-4E mutant appears to still reduce the binding between MYPT1 and MPRIP? Please explain.

6. The reviewer was concerned that the α 4E mutant appeared to still decrease the binding between MYPT1 and MPRIP (Figure 3C). The binding between MYPT1 and MPRIP was statistically analyzed in the presence of Lamtor1-Full, the Δ T1-mutant, and the α 4E mutant in Lamtor1-KO THP1 cells. The interaction did not result in a noticeable difference between Lamtor1-KO, Δ T1-mutant-, and α 4E mutant-reconstituted cells (**Response Figure 3–6**). To avoid confusion, we replaced the western blot image and added statistical information to **Figure 7d**.

Response Figure 3 – 6

7) Figure 3F - Please show extent of MPRIP knock-down in Lamtor1-KO-THP1. What about MYPT1 expression level in these cells?

7. The reviewer argued that the extent of MPRIP knockdown should be shown in Figure 3F and he/she was concerned about the expression level of MYPT1 in MPRIP knockdown cells. We neglected to show the extent of MPRIP expression in MPRIP-knockdown cells. MPRIP was knocked down (**Response Figure 3–7a**), and MYPT1 expression levels were comparable irrespective of MPRIP knockdown (**Response Figure 3–7b**). We added a western blot image to **Figure 5c**.

Response Figure 3 – 7

8) Figures 3A, B - it's interesting that the author can abolish Lamtor1 interaction with MPRIP. But is MPRIP still colocalizing with lysosome? Maybe is helpful to blot for MPRIP interaction with other lysosomal marker, or try to co-stain MPRIP with lysosome marker to examine the colocalization by using confocal imaging.

8. The reviewer was interested in knowing whether MPRIP localization to lysosomes differs in the presence and absence of Lamtor1. We compared the localization of MPRIP and LAMP1 in WT and Lamtor1^{-/-} BMDCs. The percentage of co-localization of Lamp1 and MPRIP did not differ between WT and Lamtor1^{-/-} cells (**Response Figure 3–8**). Because MPRIP binds to actin, MPRIP appears to localize independently of Lamtor1. By contrast, as shown in **Response Figure 3-2**, the localization of MYPT1 was affected by Lamtor1.

Response Figure 3 – 8

9) Figure 4B - can increased abundance of DCs itself negatively impact on mobility of DCs?

9. The reviewer asked whether increased abundance of DCs suppresses DC motility. In general, in the

experiments in which we transferred DCs into the footpad, the more DCs that were transferred, the more DCs homed to the LN. Also, the expansion of DCs due to Flt3L overexpression increases the numbers of both migratory and resident DCs (*Nat. Immunol.* 9 (6), 676–683, 2008, *J. Exp. Med.* 211 (9), 1875-189, 2014). Therefore, we do not believe that the presence of large numbers of DCs in the dermis would prevent egress from the dermis.

10) Do the authors have evidence that Lamtor1 (rather than other components of Ragulator) directly interacts with MPRIP?

10. The reviewer was interested in knowing whether Lamtor1 or other Ragulator components directly interacts with MPRIP. Lamtor1 is a string-like protein that wraps around Lamtor2–5 to form the Ragulator complex. Therefore, we assumed whether Lamtor1 directly binds with MPRIP. However, neither the C-terminal shortened form of Lamtor1 ($\Delta T1$) nor the $\alpha 4E$ -mutant Lamtor1 bound to MPRIP. Given that both mutants are unable to organize the Ragulator complex (*Nat Commun.* 2017;8(1):1625), we concluded that organization of the Ragulator complex is important for MPRIP binding.

Currently, we have not definitively identified the most important site of the Ragulator complex for binding to MPRIP. However, as mentioned earlier (**Reviewer 2–4**), the N-terminal region of MPRIP appears to be important for binding with the Ragulator complex (**Response Figure 3–9**). Additionally, the N-terminal region of MPRIP seems to associate with Lamtor1 and Lamtor2.

Response Figure 3 – 9

In the future, we will perform additional experiments, including establishment of cells harboring knockouts of MPRIP and other Ragulator components, MPRIP mutant cells, and so on, with the goal of determining how the Ragulator complex binds MPRIP.

11) Can phenotypes in Figure 4 be rescued by Lamtor1?

11. The reviewer was interested in knowing whether the impaired *in vivo* DC migration could be restored. Production of bone marrow chimeric mice in which Lamtor1 was restored in CD11c–Lamtor1^{fllox/fllox} mice was not inappropriate because of Lamtor1 is overexpressed in almost all hematopoietic cells. Therefore, we used an EGFP-expressing lentivirus vector to restore Lamtor1 to Lamtor1^{-/-} BMDCs, despite the very low transduction efficiency of this approach (approximately 8–18%). Lentivirus-exposed Lamtor1^{-/-} BMDCs, including both restored and non-restored cells, were labeled with cell-tracker violet and adoptively injected into the footpad of WT mice. Forty-eight hours later, we compared the ratio of GFP+ cell number to violet+ cell number in the popliteal LN between injected DCs and migrated DCs. The percentage of Lamtor1-restored Lamtor1^{-/-} DC (GFP+ violet+) in the popliteal LN was higher than the percentage of Lentivirus-exposed Lamtor1^{-/-} DCs (violet+), which included GFP⁺ and GFP⁻ cells (**Response Figure 3–10**). This result suggests that restoration of Lamtor1 rescues *in vivo* mobility, which was lost due to defects in Lamtor1.

Response Figure 3 – 10

We added this result to **Figure 8f** and provided the following description in the text (**page 16**).

Moreover, to determine whether adding back Lamtor1 to Lamtor1^{-/-} DCs could restore *in vivo* DC motility, we transduced Lamtor1-IRES-EGFP into Lamtor1^{-/-} BMDCs using a lentivirus system. Although the efficiency was low, approximately 7–18% in each experiment, we labeled whole BMDCs exposed to lentivirus with a cell-tracer violet dye and injected them into the footpads of WT mice. We then analyzed the abundance of GFP+ DCs, which represented Lamtor1-restored DCs, in popliteal LNs. The ratio of GFP+ DCs to violet+ DCs was higher in popliteal LNs than in the footpad (**Figure 8f**), suggesting that adding back Lamtor1 in Lamtor1^{-/-}

^{-/-} DCs restored *in vivo* DC motility.

12) Figure 4 - The authors used the subset of migratory DCs for confirming DCs trafficking and immunity adaptation. Please describe how the subset of migratory DCs were obtained from total DCs.

12. The reviewer asked about how the migratory DC subset was determined. As explained above (**Reviewer 1–6**), a very large population of CD11c⁺MHC II^{hi} cells were defined as migratory DCs. We confirmed the migratory DC population using another marker of the migratory DC subset (CD11c⁺ and CD103⁺). The number of CD11c⁺CD103⁺ DCs was also reduced in popliteal LNs of CD11c-Cre Lamtor1^{flox/flox} mice (**Response Figure 1–4**).

We added the results regarding CD103⁺DCs to **Figure 8a, b** and added the following passage to the text (**page 16**).

CD11c–Lamtor1^{flox/flox} mice had a reduced number of CD11c⁺MHCII^{hi} DCs and CD11c⁺CD103⁺DCs, both of which are representative of migratory DCs, in skin-draining LNs (dLNs) (**Figure 8a**), although they had abundant DCs in the skin itself (**Figure 8b**).

13) Does MPRIP interact with other Ragulator subunits including Lamtor2/4 and Lamtor3/5 or Rag GTPase?

13. The reviewer was interested in knowing whether MPRIP interacts with another component of the Ragulator complex, e.g., Lamtor 2, 3, 4, or 5 or RagA/C GTPases. This question is similar to a previous question (**Reviewer 3–10**). We have not yet determined the interaction site of the Ragulator complex with MPRIP; therefore, it remains unknown which component interacts MPRIP. We plan future studies to address this question, but a definitive answer will take time. I believe that our findings at this stage

are worth sharing in the literature, even though we have not yet identified the interaction sites between the Ragulator complex and MPRIP.

14) Please describe why Lamtor1 mutants ($\Delta T1$, Met1–Ser144; $\Delta T2$, Met1–Gln114; $\Delta T3$, Met1–His41; and $\alpha 4E$) were generated. Are those sites known to be interacted with Lamtor1 and other Ragulator components?

14. The reviewer asked why Lamtor1 mutants were generated. According to a previous paper (*Nat Commun.* 2017;8(1):1625), Lamtor1 consists of the following parts: the N-terminal myristoylation/palmitoylation site, which tethers Lamtor1 to the lysosome; the lysosome localization signal (Gly2–Pro25), the $\alpha 1$ helix (His41–Ser63), the $\alpha 2$ helix (His79–Val94), the $\alpha 3$ helix (Pro115–Leu119), the $\alpha 4$ helix (Phe126–Leu143), and the C-terminal tail (Arg147–Pro161). To assemble the Ragulator components into a complex, Lamtor1 binds to RagC (RD) via the Lamtor1 $\alpha 1$ helix, Lamtor3/MP-1 via the $\alpha 2$ helix, Lamtor4/p10 via the $\alpha 3$ helix, and Lamtor5/HBXIP via the $\alpha 4$ helix. Further, Lamtor1 binds to Lamtor2/p14 via the first half of the C-terminal tail (Arg147–Val156) and RagA (RD) via the second half of the C-terminal tail (Gln157–Pro161) (**Response Figure 3–11a**).

Response Figure 3 – 11

Based on this knowledge, we roughly divided Lamtor1 into four parts: the C-terminal tail, the part facing the cytoplasmic side (helices $\alpha 3$ – $\alpha 4$), the part facing the lysosome side (helices $\alpha 1$ – $\alpha 2$), and the part near the tether to the lysosome (N-terminus– $\alpha 1$ helix). Accordingly, we generated three truncated forms of Lamtor1; $\Delta T1$ (Met1–Ser144), $\Delta T2$ (Met1–Gln114), and $\Delta T3$ (Met1–His41) (**Response Figure 3–11b**). We then tried to determine which part of Lamtor1 was most important for MPRIP binding by co-immunoprecipitation with MPRIP and mutant or full-length Lamtor1.

As shown in **Supplementary Figure 5c**, none of these mutants were able to bind MPRIP, suggesting that the C-terminal tail is important. However, $\Delta T1$ is the same as CA15 in a previous report (*Nat Commun.* 2017;8(1):1625), which failed to assemble all Ragulator components, so it remained possible that complete formation of a Ragulator complex was required for binding with MPRIP. Therefore, we investigated whether the $\alpha 4E$ mutant, which could not assemble all Ragulator components, could bind to MPRIP. The results revealed that that the $\alpha 4E$ mutant could not interact

with MPRIP. In light of these findings, we concluded that the Ragulator complex is necessary for binding with MPRIP. To clarify this point, we included a schematic of the Ragulator complex in **Supplementary Figure 5a**. In addition, we described the rationale for generating truncated mutants and explained the experimental process in more detail in the text (**page 14**) as follows.

Thus, to determine the regions of Lamtor1 that are critical for assembly of the Ragulator complex formation and interaction with MPRIP, we divided Lamtor1 into four parts based on the structure of the Ragulator complex: the C-terminal tail of Lamtor1, the part facing the cytoplasm side ($\alpha 3$ – $\alpha 4$ helix), the part facing the lysosome side ($\alpha 1$ – $\alpha 2$ helix), and the part near the stalk to anchor the Ragulator complex to the lysosome (the N-terminal– $\alpha 1$ helix) (**Supplementary Figure 5a**). We generated three FLAG-tagged truncated forms of Lamtor1; $\Delta T1$ (Met1–Ser144), $\Delta T2$ (Met1–Gln114), and $\Delta T3$ (Met1–His41) (**Supplementary Figure 5b**). All truncated Lamtor1 variants failed to form the Ragulator complex (**Supplementary Figure 5c**), confirming that the C-terminal tail of Lamtor1 is essential for Ragulator complex formation. When we introduced the $\Delta T1$ -mutant into MPRIP-V5-HEK293T cells, the interaction of MPRIP with $\Delta T1$ mutant was completely abolished (**Figure 7a, b**).

15) What is % input in chemotaxis experiments evaluated by Transwell assay?

15. The reviewer requested that we report the percent input in the chemotaxis assay. In the chemotaxis assay using Transwell, approximately 10^5 cells were applied to the upper chamber, and the number of cells in the lower chamber was counted after 3–4 hours of chemotaxis. Migration ability (= % of input cells) was evaluated by dividing the number of cells in the lower chamber by the number of cells input into the upper chamber and multiplying by 100. We described this more clearly in the Materials and Methods (**page 40**).

Minor:

1) Western blot data is needed to verify over expression, knockdown or knockout. For example, Figure 2F: blot for Lamtor1, Figure 3F: blot for MPRIP, etc.

1. The Reviewer pointed out the lack of western blot images showing Lamtor1 overexpression in Figure 2f and MPRIP knockdown in Figure 3f. According to this comment, we added Lamtor1 band images to **Figure 5a** and **Figure 5c**.

2) line 87: “Hence, we focused on the Ragulator complex because of the similar distribution ...”. The rationale

of focusing on Ragulator is very farfetched, because any lysosomal protein will have similar distribution here.

2. The reviewer pointed out that the rationale for focusing on Lamtor1 based on the similarity of distribution was exaggerated. As the reviewer pointed out, it may be a leap to focus on Lamtor1, as there are many other lysosome proteins. Lamtor1-deficient MEFs exhibit abnormal localization of lysosomes and disrupted recycling of β 1-integrin. In mesenchymal cells, such as fibroblasts, detachment of the trailing edge from the ECM mediated by integrin recycling is important for migration, suggesting that Lamtor1 is involved in mesenchymal cell motility. On the other hand, movement of amoeboid-like cells, including DCs, does not involve integrins to a large extent, especially in a three-dimensional environment, whereas lysosomes play roles in DC migration. This is why we were interested in Lamtor1 and expected that Lamtor1 on lysosomes would regulate cell motility in an integrin-dependent manner. We revised the introduction to make it easier to understand why we focused on Lamtor1 (**page 5**) and inserted the following sentence into the text (**page 8**).

Given that Ragulator complex expressed on the lysosomal membrane is involved in cell movement, we focused on Lamtor1 and examined the localization of Lamtor1 in polarized and non-polarized cells.

3) Figures 1B, 1C - What is a stimulator for polarized cells?

3. The reviewer asked which stimulus was used to induce polarization in Figure 1b, c. No stimulant was used to induce polarization in Figures 1b and 1c because MLC phosphorylation in Lamtor1^{-/-} DCs was reduced with or without chemokine stimulation. Polarized and non-polarized cells were distinguished based on cell morphology. We defined polarized cells in Materials and Methods (**page 42**) and the legend of Figure 1b and 1c as follows:

Cells were classified as polarized or non-polarized based on morphology.

4) Figure 1D legend - "the indicated concentrations of CCL19 (left) or CCL21 (right) in a Transwell assay system" should be changed to match the arrangement of figure.

4. The reviewers pointed out the improper order in Figure 1d. We corrected the figure legend.

5) Figure 1J - Authors showed the expression of p-MLC in Lamtor1 WT and KO of DC cells. What expression level of p-MLC when those cells (Lamtor1 WT and KO) exposed in CCL19 or CCL21?

5. The reviewer questioned the expression level of p-MLC when exposed to CCL19 or CCL21. As mentioned above (**Reviewer 1–4**), phosphorylation of MLCs in WT DCs was comparable with or without CCL19 administration (**Response Figure 3–12**).

Response Figure 3 – 12

6) Figure 1J, show the blots of Lamtor1 knockout and LAMP1 level.

6. The reviewer requested that we add western blot images for Lamtor1 and Lamp1 to Figure 1j. We thank the reviewer for pointing out our mistake. We added a Lamtor1 blot to Figure 1J; Lamp1 expression did not differ significantly between WT and Lamtor1^{-/-} DCs (**Response Figure 3-13b**).

Response Figure 3 – 13

We added these data, with statistical information, to **Figure 2g** and **2i**.

7) Figure 2D use endogenous IP to demonstrate Lamtor1 and MPRIP interaction in a physiological relevance.

7. The reviewer asked whether endogenous Lamtor1 and MPRIP could be co-immunoprecipitated. We examined co-precipitation of Lamtor1 and MPRIP in primary BMDCs and THP1 cells. However, it was difficult to detect co-precipitation of Lamtor1 and MPRIP, probably due to the low affinity of the antibodies and low endogenous expression levels of these proteins.

8) Figure 2G, show blots of Lamtor1. Figure 3d, show Lamtor1-FLAG and α 4E-FLAG expression. Figure 3f, show MPRIP expression level.

8. The reviewer pointed out the lack of blots for Lamtor1, Lamtor1-FLAG, and α 4E-Flag in Figure 2g, 3d, and for MPRIP in Figure 3f. We thank the reviewer for pointing out our mistakes. We add blots for Lamtor1 to these Figures (**Response Figure 3-14**). The expression level of MPRIP in Figure 3f is mentioned in **Response to reviewer 3-7**.

Response Figure 3 - 14

9) Draw out a working model.

9. The reviewer requested to draw the working model. To help readers understand our findings, we included a graphical summary of this article in **Figure 10** and explained our working model in Discussion as follows (**page 20**).

Response Figure 3 - 15

In this study, we revealed the novel significance of peripherally transported lysosomes in motile cells, in which the Ragulator complex inactivates MLCP by binding to MPRIP. This binding interferes with the interaction between MPRIP and MYPT1 and ultimately facilitates myosin II activity. That is, in immotile cells, the Ragulator complex localized to lysosomes distributed preferentially in the perinuclear region, and MPRIP anchors MLCP on myosin–actin bundles by binding MYPT1, a subunit of MLCP, resulting in suppression of MLC phosphorylation. In motile cells, lysosomes bearing the Ragulator complex move to the uropod, where the Ragulator complex binds to MPRIP. This binding interferes with the interaction between MPRIP and MYPT1 and decreases MLC phosphatase activity, thereby increasing MLC phosphorylation. Consequently, cell motility is facilitated (**Figure 10**).

REVIEWERS' COMMENTS

Reviewer #1 (Remarks to the Author):

The authors have responded well to my open questions. I do not have any further concerns.

Reviewer #2 (Remarks to the Author):

The manuscript is much improved, thorough, and more clear to read and understand. Two minor points are noted:

- 1) The description of migration mechanisms in fibroblasts in the introduction seems long with respect to its relevance to the paper, since this is not the type of motility under study.
- 2) In the graphical abstract, the representation of MLC as a hemicircle seems odd. MLC is part of myosin, which is generally represented as a dimer with the heads and tails that indicate how it functions as a motor, with the head binding actin fibers.

Changing these descriptions would further make the manuscript clear for readers familiar with these other fields but new to immunology.

Reviewer #3 (Remarks to the Author):

All concerns except point 13 have been addressed.

The authors could have easily looked at LAMTOR 4, LAMTOR 5, and Rag GTPase binding in respect to the MPRIP.

Point to point Replay

Reviewer #2 (Remarks to the Author):

1) The description of migration mechanisms in fibroblasts in the introduction seems long with respect to its relevance to the paper, since this is not the type of motility under study.

Thank you for the comment. According to the reviewer's suggestion, we deleted the explanation the mode of mesenchymal cell migration.

2) In the graphical abstract, the representation of MLC as a hemicircle seems odd. MLC is part of myosin, which is generally represented as a dimer with the heads and tails that indicate how it functions as a motor, with the head binding actin fibers.

Thank you for pointing out the inaccuracy. We improved illustrations to accurately show the relevance of myosin light chains, myosin heavy chain, and actin fibers (**Response Figure 1**).

Reviewer #3 (Remarks to the Author):

All concerns except point 13 have been addressed. The authors could have easily looked at LAMTOR 4, LAMTOR 5, and Rag GTPase binding in respect to the MPRIP.

Thank you for the comment. We apologize for not displaying the data at that time. No interaction between MPRIP and LAMTOR4 or LAMTOR5 occurred (**Response Figure 2**), and the interaction between MPRIP and Rag GTPases was not tested.

Response Figure 2